

# Taxonomic revision of black salamanders of the *Aneides flavipunctatus* complex (Caudata: Plethodontidae)

Sean B. Reilly[1,2,*] and David B. Wake[1,*]

[1] Museum of Vertebrate Zoology and Department of Integrative Biology, University of California, Berkeley, CA, United States of America

[2] Department of Ecology and Evolutionary Biology, University of California, Santa Cruz, CA, United States of America

[*] These authors contributed equally to this work.

Corresponding author
Sean B. Reilly, sreilly@ucsc.edu

## ABSTRACT

We present a taxonomic revision of the black salamander (*Aneides flavipunctatus*) complex of northwestern California and extreme southeastern Oregon. The revision is based on a number of published works as well as new molecular and morphological data presented herein. The subspecies *Aneides flavipunctatus niger* Myers & Maslin 1948 is raised in rank to a full species. It is isolated far to the south of the main range on the San Francisco Peninsula, south and west of San Francisco Bay. Another geographically isolated set of populations occurs well inland in Shasta County, northern CA, mainly in the vicinity of Shasta Lake. It is raised from synonymy and recognized as *Aneides iecanus* (Cope 1883). The remaining taxa occur mainly along and inland from the coast from the vicinity of the Russian River and Lake Berryessa/Putah Creek, north to the vicinity of the Smith River near the Oregon border and more inland along the Klamath and Trinity Rivers and tributaries into Oregon. The northern segment of this nearly continuous range is named *Aneides klamathensis* Reilly and Wake 2019. We use molecular data to provide a detailed examination of a narrow contact zone between the northern *A. klamathensis* and the more southern *A. flavipunctatus* in southern Humboldt County in the vicinity of the Van Duzen and main fork of the Eel rivers. To the south is the remnant of the former species and it takes the name *Aneides flavipunctatus* (Strauch 1870). It is highly diversified morphologically and genetically and requires additional study.

## INTRODUCTION

The black salamander, *Aneides flavipunctatus*, occurs in the coastal forests and mountains of northwestern California and extreme southwestern Oregon. In recent decades the extent of the geographic range has been refined and extended due to survey work, especially in the northern extent of the range in Oregon (*Olson, 2008*; *Reilly et al., 2013*). It has long been recognized that this species contains striking regional variation in color pattern (Fig. 1) and microhabitat preference (*Lowe, 1950*; *Lynch, 1981*), and since the advent of molecular genetics a number of studies have attempted to use genetic variation to understand and

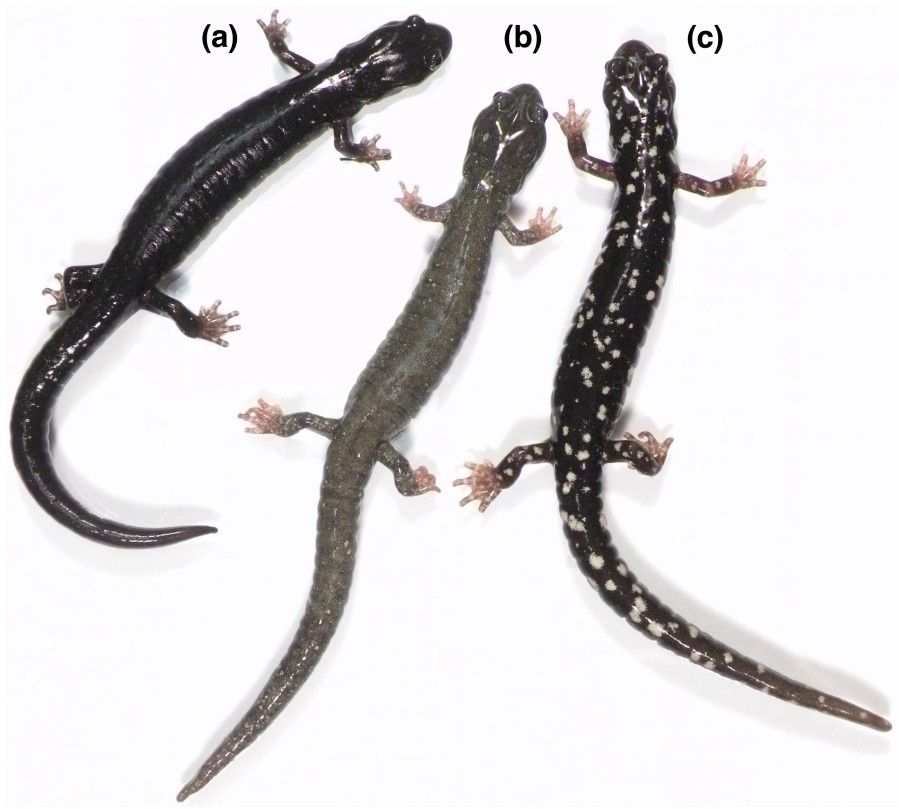

**Figure 1  Three contrasting color morphologies found within the *Aneides flavipunctatus* complex.**
(A) The pure black morph is characteristic of *Aneides niger* and southern coastal populations of *Aneides flavipunctatus*, and this individual is from near Fort Bragg, Mendocino Co., CA. (B) The frosted morph is characteristic of *Aneides klamathensis* and can also be found in northern populations of *Aneides flavipunctatus*; this individual is from near Scotia, Humboldt Co. CA (population 14 in Fig. 2). (C) The spotted morph is characteristic of *Aneides iecanus* (although with smaller and more numerous spots than in this specimen) and southern inland populations of *Aneides flavipunctatus*; this individual was found near Boonville, Mendocino Co., CA. All of these salamanders are from the range of the revised *A. flavipunctatus,* illustrating the high degree of color pattern variation present within the species (photo: D Portik).

explain this eco-morphological variation (*Larson, 1980*; *Rissler & Apodaca, 2007*; *Reilly, Marks & Jennings, 2012*; *Reilly et al., 2013*; *Reilly & Wake, 2015*). While many researchers familiar with the species suggested that *Aneides flavipunctatus* is a multispecies complex, the mosaic of genetic, ecological, and morphological patterns across its geographic range raises questions concerning diagnosability and boundaries of putative species.

The types of *Aneides flavipunctatus* (*Strauch, 1870*) were collected by the Russian biologist IG Voznesenskii, most likely in 1841 (see below). The listed type locality is "Californien (Neu-Albion)" (*Strauch, 1870*). New Albion (English version) was a general term for that part of coastal California north of San Francisco Bay that was unoccupied by Mexico, where the Russian colony at Fort Ross had been founded in 1812.

*Cope (1883)* found a single subadult salamander from the vicinity of the federal fish hatchery on the McCloud River (close to its juncture with the Pit River, currently under the

water of Lake Shasta), in present-day Shasta County, California, and named it *Plethodon iecanus*. Once he had seen an adult *Cope (1886)* recognized that his taxon was a relative of *Aneides lugubris* (*Hallowell, 1849*) and *Aneides ferreus Cope, 1869*. However, he did not know *P. flavipunctatus* and in his monograph of North American Batrachia (present-day Amphibia) (*Cope, 1889*) left that taxon in *Plethodon*, where it remained until *Storer (1925)* recognized it as an *Aneides* and reduced *A. iecanus* to its synonymy. The most recently named member of what we here consider the *Aneides flavipunctatus* complex is *Aneides flavipunctatus niger Myers & Maslin, 1948*, a taxon based on populations from the Santa Cruz Mountains on the southern San Francisco Peninsula that has distinctive solid black coloration in adults, with almost no white spotting. The disjunct distribution is well to the south of the rest of the range of the complex. Populations north of San Francisco Bay (including those from inland Shasta County) were all considered to be *A. f. flavipunctatus* (*Myers & Maslin, 1948*). Shortly afterwards, an unpublished Ph.D. dissertation by *Lowe (1950)* recognized five subspecies within the complex based on color pattern and microhabitat preference: *A. f. niger* from the Santa Cruz Mountains, *A. f. iecanus* from Shasta County, *A. f.* "sequoiensis" from the Klamath Mountains south to northern Mendocino County near Laytonville, *A. f.* "quercetorum" from inland areas south of Laytonville, and *A. f. flavipunctatus* from coastal areas south of Laytonville. Despite not being formally published, meaning that his new names lacked validity, Lowe's dissertation was influential and suggested some of the morphological and ecological variation present within the species. Furthermore, it identified a region in central Mendocino County where multiple ecomorphs were thought to be parapatrically distributed.

*Larson (1980)* reported data for 21 allozyme loci from 22 populations and suggested that the complex experienced a nearly simultaneous distributional fragmentation throughout its range during the Pleistocene. However, some populations (notably those on the southern San Francisco Peninsula and in inland Shasta County) were well differentiated from each other and from populations in the main body of the range. Around this same time a detailed field and lab study of geographic variation throughout the range of the entire complex by *Lynch (1981)* found, as in *Lowe*'s (*1950*) more limited study, significant geographic variation in morphology and color pattern. Although many populations had discrete, densely pigmented iridophores, Lynch found that all aspects of coloration in *A. flavipunctatus* showed ontogenetic variation. Across the range juveniles displayed brassy pigmentation caused by embedded iridophores overlain by xanthophore pigments. Lynch hypothesized that *A. flavipunctatus* was formerly more widespread and morphologically uniform, but that their range became fragmented due to climatic fluctuations, and that isolated populations began to simultaneously differentiate in response to local environmental conditions as a result of the high topographic and climatic diversity in northwestern California. This process ultimately increased the genetic, morphological, and ecological diversity within *A. flavipunctatus,* leading to divergent lineages that now exhibit highly variable rates and directions of ontogenetic-based morphological evolution (*Lynch, 1981*). Lynch recommended that a single species be recognized, with no subspecies.

More recent work documented molecular diversification (mitochondrial (mt) DNA sequence data) (*Rissler & Apodaca, 2007*), and sampling of genes (nuclear DNA sequence

data) and localities expanded greatly (*Reilly, Marks & Jennings, 2012*; *Reilly et al., 2013*; *Reilly & Wake, 2015*). DNA sequence data for 2 mtDNA loci from 18 localities (42 samples) were presented by *Rissler & Apodaca (2007)*, who identified four major mitochondrial clades (identified as Central, Northwest, Shasta and Southern Disjunct). The sampling was relatively sparse and did not permit identification of boundaries between clades or determination of the degree of genetic isolation of those clades. Using rough range estimates from their mtDNA phylogeny, Rissler & Apodaca conducted an ecological niche analysis and a contact zone suitability test, finding that each clade contained distinct, yet overlapping, niches. The limited sampling forced the authors to assume the clade designation for many hundreds of locality records using rough geographic criteria. For example, the predicted contact zone for their Central and Northwest clades (corresponding to the Central Core and Northwest clades, respectively, of *Reilly & Wake, 2015*; i.e., the taxa *A. flavipunctatus* and *A. klamathensis,* respectively, of this study) was approximately 40 km south of the actual contact zone. Their suitability analysis found their proposed contact zone to be unsuitable for the Central clade, but suitable for the Northwest clade (which in fact ends well to the north). While these types of niche models and contact zone analyses can provide insights into ecologically driven lineage divergence, such studies are only as reliable as the lineage boundaries, which requires accurate assignment of an adequate number of populations.

*Rissler & Apodaca (2007)* made no formal taxonomic changes, but they did recommend a four species taxonomy. The ill-defined boundary between the Central and Northwest clades (*A. flavipunctatus* and *A. klamathensis* sp. nov.) and the level of reproductive isolation between them were resolved by *Reilly & Wake (2015)*, who sequenced 3 mtDNA loci for 240 samples from 136 localities, along with 13 nDNA loci for 145 samples from 93 localities. *Reilly & Wake (2015)* confirmed the existence of four major genetic lineages within the complex, and within the contiguous main range a boundary between northern and southern taxa lies along and just south of the Van Duzen River and tributaries in southern Humboldt County, CA. Gene flow estimates across this contact zone were both well below the 2*Nm* value of 1, suggesting that there has not been sufficient gene exchange to prevent species divergence.

In this study we (1) compile the findings of all previous work on *Aneides flavipunctatus*, (2) provide new genetic data that better defines the Humboldt County contact zone, (3) utilize morphometric data to distinguish putative species, and (4) formally recognize four distinct species-level taxa within the complex. The geographically disjunct subspecies *Aneides flavipunctatus niger* Myers & Maslin 1948 is elevated to *Aneides niger*, *Plethodon iecanus* (Cope 1883) is recognized as *Aneides iecanus*, a new taxon represented by populations north of the Van Duzen River region (see below and *Reilly & Wake, 2015* for a full range description) is named *Aneides klamathensis*, and populations in the coast ranges south of the Van Duzen River region to Sonoma and Napa Counties (originally *Plethodon flavipunctatus* Strauch 1870) retain the name *Aneides flavipunctatus*. Highly divergent lineages remain within the revised *Aneides flavipunctatus*, and we suggest that next-generation sequencing along with dense sampling will be needed to determine the number of independently evolving lineages.

## MATERIALS AND METHODS

### Taxon sampling

We collected 18 black salamanders from nine localities in inland regions of southeastern Humboldt County at the contact zone between *A. flavipunctatus* and *A. klamathensis* sp. nov. in order to pinpoint the boundary between these species (Table 1). The sampling transect follows Alderpoint Road between the Eel River at Alderpoint north through Bridgeville and into the Van Duzen drainage. All samples are deposited at the Museum of Vertebrate Zoology, UC Berkeley.

Animal use was approved by the University of California, Berkeley, IACUC protocol #R093-0205 issued to DBW. Collection of live salamanders in the field was authorized by the California Natural Resources Agency, Department of Fish and Wildlife (approval# SC-2860 issued to DBW).

### Genetic analysis

We sequenced a portion of the mitochondrial *ND4* gene from the salamanders collected in SE Humboldt County following methods described in *Reilly & Wake (2015)*. These sequences were combined using GENEIOUS (*Kearse et al., 2012*) and aligned to existing mitochondrial sequences (*ND4, 12S,* and cytb) for *Aneides flavipunctatus* used in previous studies (*Rissler & Apodaca, 2007*; *Reilly, Marks & Jennings, 2012*; *Reilly et al., 2013*; *Reilly & Wake, 2015*) using MUSCLE (*Edgar, 2004*).

Phylogenetic analysis was performed using RAxML (*Stamatakis, 2014*) in order to place new samples into known mtDNA clades described in *Reilly & Wake (2015)*. The best fit model of sequence evolution (GTR + I + G) was determined using jModelTest (*Posada, 2008*), and 1,000 bootstrap replicates were performed to evaluate nodal support. The tree was rooted with *Aneides lugubris, A. vagrans, A. ferreus,* and *A. hardii* sequences used in *Reilly & Wake (2015)*.

### Morphological analyses

We measured morphological characters to the nearest 0.1 mm from adult *Aneides flavipunctatus* specimens representing the species described in this paper. We used the Northern "Central Core" clade from *Reilly & Wake (2015)* to represent *A. flavipunctatus* because we are most interested in the change in morphology close to the southern Humboldt contact zone. Males and females were analyzed separately with measurements from 10–11 adults per population for *A. niger*, *A. iecanus*, *A. klamathensis,* and North Central Core *A. flavipunctatus.* Measurements taken include snout to the posterior margin of the vent (SVL), axilla to groin (AG), head width (HW), forelimb length (FLL), hind limb length (HLL), right hand (RH), right foot (RF), longest toe (LT), distance between eyes (DBE), internarial distance (ID), and snout to gular fold (SG) (see *Bingham et al., 2018*). Morphological analyses were calculated using the MASS package (*Venables & Ripley, 2002*) implemented in R (*R Core Team, 2018*) to differentiate the four species. All measurements were log transformed before conducting a linear discriminant function analysis (DFA).

Osteological descriptions of each species are presented based on the examination of cleared and stained specimens housed at the Museum of Vertebrate Zoology. The number

**Table 1**  Voucher numbers and localities for newly collected Aneides samples from southeastern Humboldt County. The locality numbers refer to Fig. 2.

| Museum # | Collector # | Sex | County | Lat | Long | Haplotype group | Locality # |
|---|---|---|---|---|---|---|---|
| MVZ:Herp:272722 | DBW 6645 | F | Humboldt | 40.19155 | −123.58872 | *A. flavipunctatus* | 3 |
| MVZ:Herp:272723 | DBW 6646 | juv | Humboldt | 40.19155 | −123.58872 | *A. flavipunctatus* | 3 |
| MVZ:Herp:272724 | DBW 6647 | F | Humboldt | 40.19155 | −123.58872 | *A. flavipunctatus* | 3 |
| MVZ:Herp:272725 | DBW 6648 | M | Humboldt | 40.20833 | −123.59608 | *A. flavipunctatus* | 5 |
| MVZ:Herp:272726 | DBW 6649 | M | Humboldt | 40.22648 | −123.60881 | *A. flavipunctatus* | 6 |
| MVZ:Herp:272727 | DBW 6650 | M | Humboldt | 40.22648 | −123.60881 | *A. flavipunctatus* | 6 |
| MVZ:Herp:272728 | DBW 6651 | F | Humboldt | 40.26458 | −123.62753 | *A. klamathensis* sp. nov. | 7 |
| MVZ:Herp:272729 | DBW 6652 | M | Humboldt | 40.26458 | −123.62753 | *A. klamathensis* sp. nov. | 7 |
| MVZ:Herp:272730 | DBW 6653 | F | Humboldt | 40.26458 | −123.62753 | *A. klamathensis* sp. nov. | 7 |
| MVZ:Herp:272731 | DBW 6654 | F | Humboldt | 40.30816 | −123.65466 | *A. klamathensis* sp. nov. | 8 |
| MVZ:Herp:272734 | DBW 6657 | M | Humboldt | 40.33452 | −123.67963 | *A. klamathensis* sp. nov. | 9 |
| MVZ:Herp:272735 | DBW 6658 | M | Humboldt | 40.33452 | −123.67963 | *A. klamathensis* sp. nov. | 9 |
| MVZ:Herp:272736 | DBW 6659 | M | Humboldt | 40.33452 | −123.67963 | *A. klamathensis* sp. nov. | 9 |
| MVZ:Herp:272737 | DBW 6660 | juv | Humboldt | 40.33452 | −123.67963 | *A. klamathensis* sp. nov. | 9 |
| MVZ:Herp:272738 | DBW 6661 | M | Humboldt | 40.33452 | −123.67963 | *A. klamathensis* sp. nov. | 9 |
| MVZ:Herp:272740 | DBW 6663 | F | Humboldt | 40.34788 | −123.71286 | *A. klamathensis* sp. nov. | 10 |
| MVZ:Herp:272741 | DBW 6664 | juv | Humboldt | 40.38870 | −123.74288 | *A. klamathensis* sp. nov. | 11 |
| MVZ:Herp:272742 | DBW 6665 | M | Humboldt | 40.41800 | −123.76150 | *A. klamathensis* sp. nov. | 12 |

of specimens, their sex, age class, and museum numbers are presented in the osteological descriptions for each species.

## Nomenclatural acts

The electronic version of this article in Portable Document Format (PDF) will represent a published work according to the International Commission on Zoological Nomenclature (ICZN), and hence the new names contained in the electronic version are effectively published under that Code from the electronic edition alone. This published work and the nomenclatural acts it contains have been registered in ZooBank, the online registration system for the ICZN. The ZooBank LSIDs (Life Science Identifiers) can be resolved and the associated information viewed through any standard web browser by appending the LSID to the prefix http://zoobank.org/. The LSID for this publication is: urn:lsid:zoobank.org:pub:D11721DC-3000-4EA6-BC51-87D47D3277CB. The online version of this work is archived and available from the following digital repositories: PeerJ, PubMed Central and CLOCKSS.

## RESULTS

### Contact zone sampling and genetic analysis

Sample localities and their associated species assignment can be viewed in Fig. 2 with numbers alongside relevant localities. Mitochondrial *ND4* haplotypes for six of the newly sequenced salamanders from the three southern-most localities belonged to *A. flavipunctatus,* while haplotypes for 12 salamanders from the six more northern localities

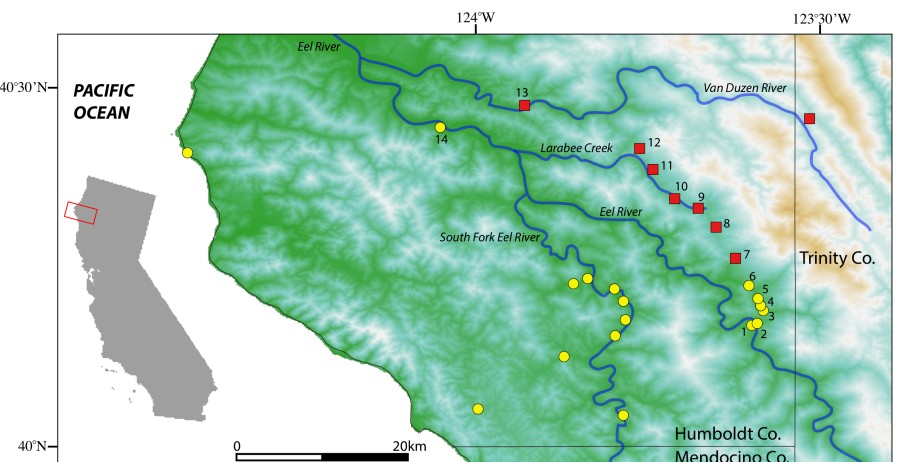

**Figure 2 Map of sample localities from the southern Humboldt County contact zone.** *Aneides flavipunctatus*, yellow circles; *Aneides klamathensis*, red squares. Greener shades represent lower elevations and white to brown colors represent higher elevations. Locality numbers are for new samples in this study (see Table 1).

belonged to *A. klamathensis* (Fig. 2). These haplotypes were grouped within these clades with high bootstrap support (Fig. 3). The two haplotype groups are approximately 5.4% divergent (*Reilly & Wake, 2015*). Locality numbers of samples within the tree are noted by the bold number after the museum number. This narrows the contact zone in eastern Humboldt County between *A. flavipunctatus* and *A. klamathensis* to a ~3 km region between Dobbyn Creek and the town of Blocksburg.

## Morphological analyses

Measurements can be found in Table S1. For the male-specific LDA analysis 0.686 of the variation was captured in the first linear discriminant, 0.192 in the second, and 0.122 in the third. In the female-specific analysis 0.504 of the variation was captured in the first linear discriminant, 0.356 in the second, and 0.140 in the third. Plotting of linear discriminant 1 vs 2 for both males and females can be found in Fig. 4. Attempts at using the DFA to classify adult salamanders from the southern Humboldt contact zone were relatively unsuccessful, suggesting a lack of morphological distinctiveness between *A. flavipunctatus* and *A. klamathensis* along the Alderpoint Road contact zone.

## Geographic range estimation

The geographical ranges of the four species have been estimated based on previous genetic studies as well as our judgement and experience in the field. All museum localities have been downloaded from VertNet (vertnet.org) and plotted according to species designations (Fig. 5). In some cases, such as the records from western Tehama and western Glenn Counties, even examination of specimens and collector field notes were not sufficient to assign them to a species at this time.

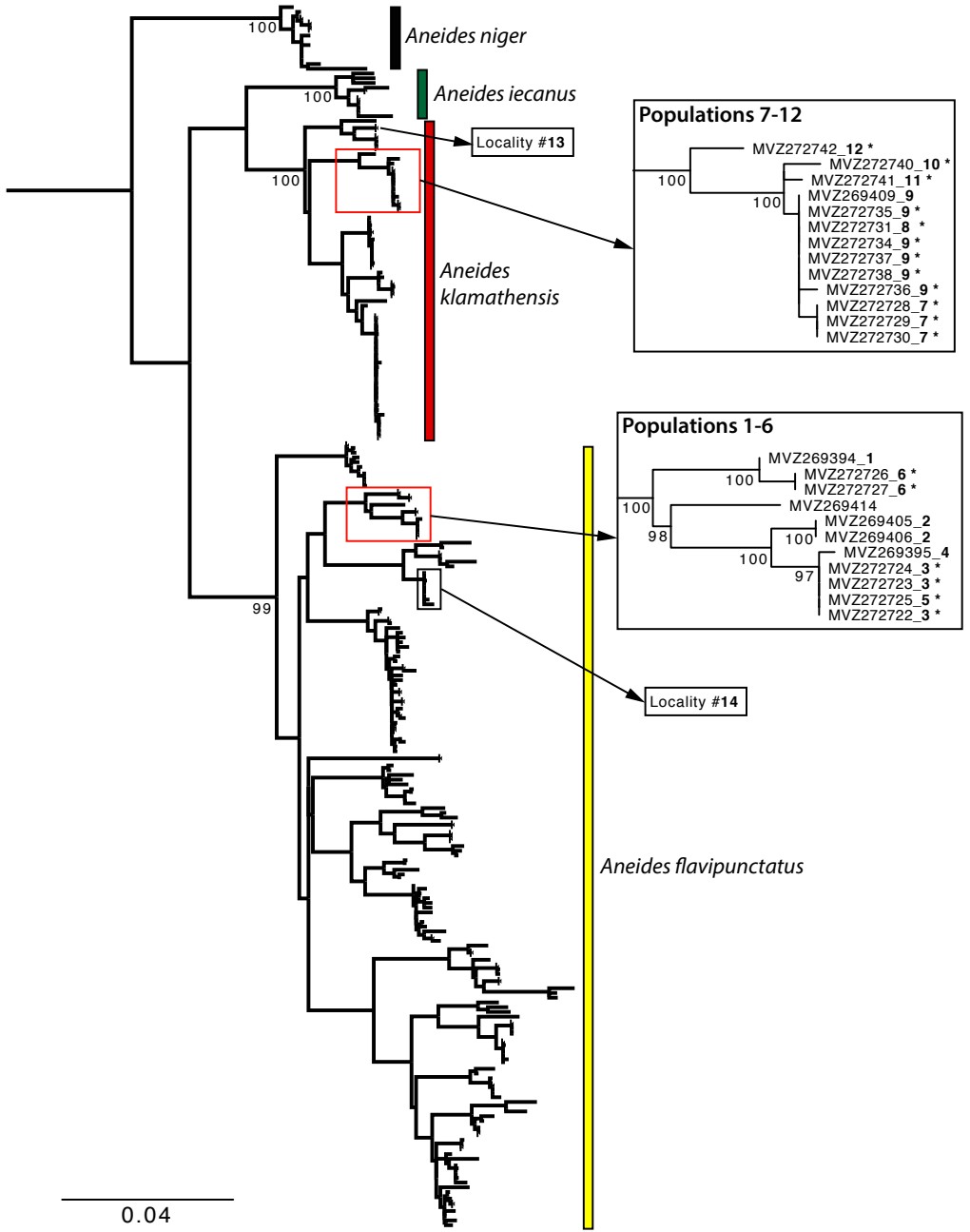

**Figure 3** **Maximum Likelihood phylogeny of the mitochondrial *ND4* gene.** Newly sampled sequences from the Southern Humboldt contact zone are denoted by an asterisk. Bold numbers after the specimen number refer to locality numbers in Fig. 2. Localities 13 and 14 are nearby sites for *A. klamathensis* and *A. flavipunctatus,* respectively.

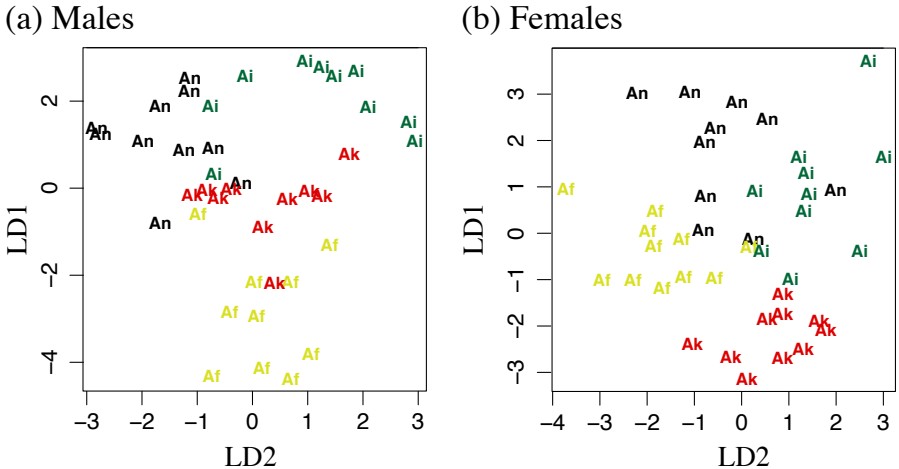

(a) Males          (b) Females

**Figure 4   Linear discriminant function analyses of 11 log transformed morphometric measurements.**
LD1 plotted against LD2 for (A) males and (B) females. Af, *Aneides flavipunctatus*; Ak, *Aneides klamathensis*; Ai, *Aneides iecanus*; An, *Aneides niger*.

# SYSTEMATICS

## *Aneides klamathensis*, new species

*Aneides flavipunctatus* (part)—*Stebbins, 1951*
*Aneides* sequoiensis (part; *nomen nudum*)—Dubois and Raffaëlli, 2009
Klamath Black Salamander
Figs. 6–7

*Holotype:* MVZ 291759 (Museum of Vertebrate Zoology, University of California, Berkeley, California, USA) (Field number: SBR 265), an adult male from ~1 km east of Klamath, Del Norte County, California (coordinates 41.52213 N, 123.99712 W; error ±9 m; elevation 20 m), collected by SB Reilly and DB Wake on 10 November 2013.

*Paratypes:* MVZ 217468 (Female, F), 217469 (Male, M), East Fork Rd, 1 mi (1.6 km) N Hwy 299 at Helena, Trinity Co, CA (40.7825366 N, 123.1283051 W); MVZ 124234 (M), 124236 (F), 124237 (F), Big Slide Campground, 7.2 mi (11.6 km) N (rd) Hyampom, Trinity Co, CA (40.6792098 N, 123.5086485 W); MVZ 184687 (F, cleared and stained), 12.6 mi (20.3 km) N Hyampom, Humboldt Co, CA (40.7217669 N, 123.5516924 W); (MVZ 124229 (M), 124230 (F), Hwy 36, 0.3 mi (0.48 km) W Mad River (town), Trinity Co, CA (40.4545945 N, 125.5103053 W); MVZ 196362 (M), 10.2 mi (16.4 km) E Big Bar, Trinity Co, CA (40.7622585 N, 123.0983789 W); MVZ 124223 (F), 2.8 mi (4.5 km) E (rd) Hawkins Bar, along state hwy 299, Trinity Co., CA (40.8497268 N, 123.4848539 W); MVZ 124220 (F), 124221 (F), MVZ 124222 (M), 0.5 mi (0.8 km) E Junction City on hwy 299, Trinity Co, CA (40.7278373 N, 123.0400778 W); MVZ 124225 (F), Little Bidden Creek, 1.4 mi (2.25 km) S and 1.3 mi (2.1 km) E Burnt Ranch, Trinity Co, CA (40.783976 N, 123.458849

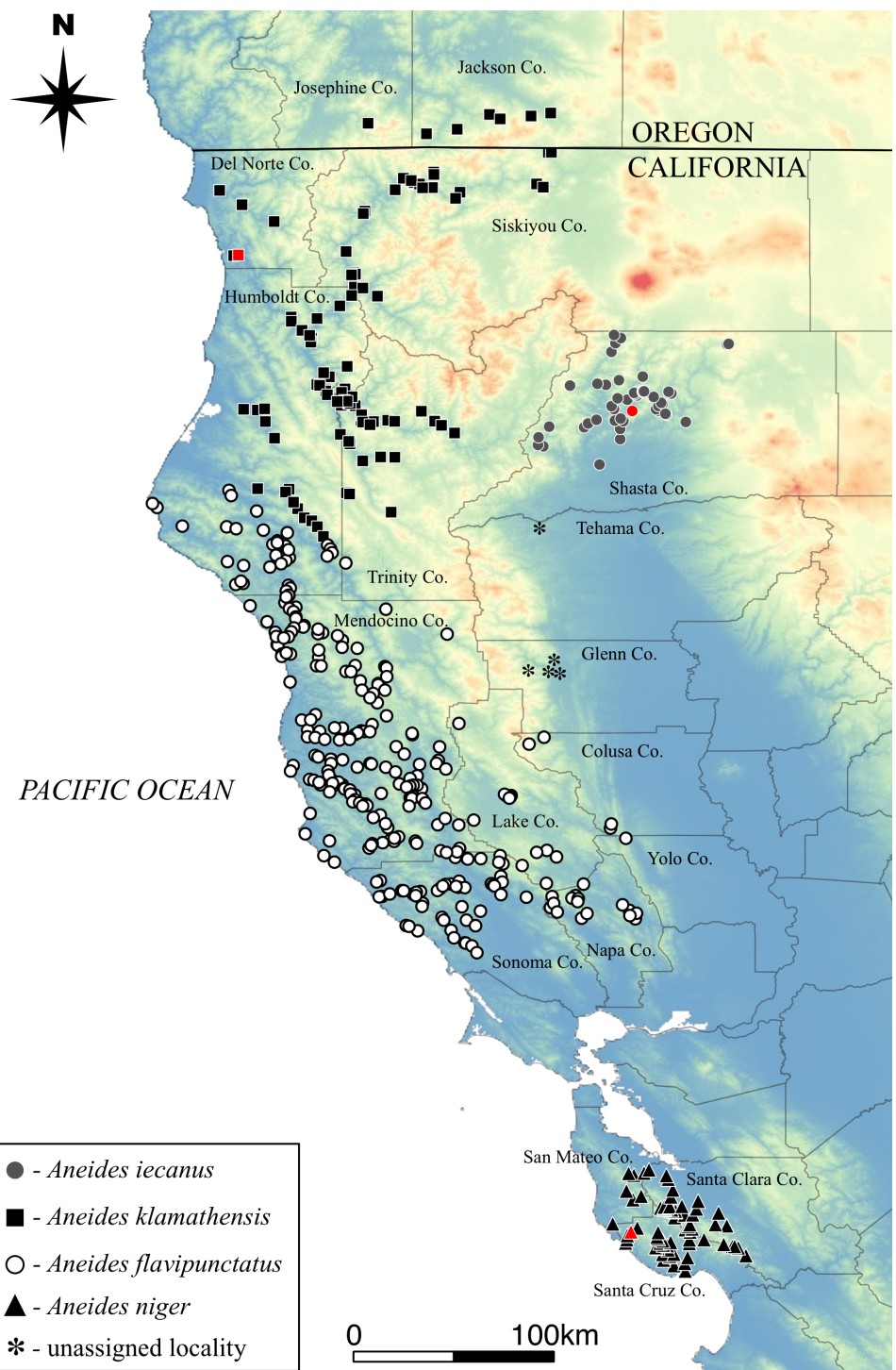

**Figure 5** **Distribution of the four species comprising the *Aneides flavipunctatus* complex.** Known type localities for three species shown in red symbol. Type locality of *A. flavipunctatus* likely in northern Sonoma Co. (see text for detailed 'Discussion'). Asterisks represent black salamander localities that are of unknown species origin.

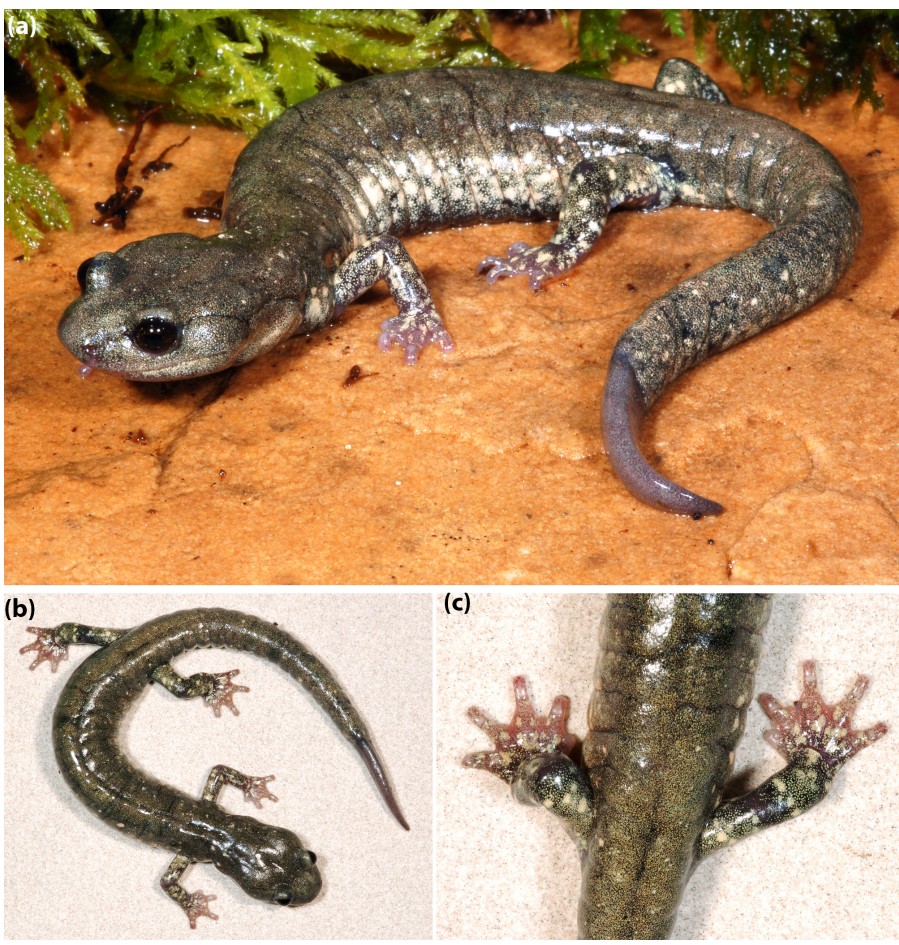

**Figure 6** **Holotype of *Aneides klamathensis*, MVZ 291759, and adult male, photographed in life.** (A) View from the side, (B) dorsal view showing the solid black ground color overlain by greenish-gray pigment that extends partially down the lateral flank of the trunk, and (C) closeup of the rear showing the scattered cream-colored spots that are most numerous on the limbs but relatively few on other dorsal surfaces (photos: M Mulks).

W); MVZ 124238 (M), state hwy 299 3.5 mi E (rd) Salyer, Trinity Co, CA (40.8739495 N, 123.5337218 W); MVZ 184682 (M, cleared and stained), state hwy 299 2 mi (3.2 km) E Salyer, Trinity Co, CA (40.8832655 N, 123.5545419 E); MVZ 217462 (F), rest area 2.5 mi (4 km) SE Salyer, Trinity Co, CA (40.8832291 N, 123.5469397 W); MVZ 217471 (F), 12.2 mi (19.6 km) SE Salyer, Trinity Co, CA (40.76509 N, 123.42883 W); MVZ 196349 (M), hwy 96, 5.5 mi (8.9 km) S (rd) Weitchpec, Humboldt Co, CA (41.1252012 N, 123.6837375 W); MVZ 124006 (M), 1 mi (1.6 km) N Weitchpec, Humboldt Co, CA (41.18762 N, 123.72346 W); MVZ 199753 (F, cleared and stained), 4.7 mi (5.6 km) S Weitchpec, Humboldt Co, CA (41.1346733 N, 123.6838263 W); MVZ 124011 (M), 4.9 mi (7.9 km) S Weitchpec on hwy 96, Humboldt Co, CA (41.1326067 N, 123.6863681 W), MVZ 221018 (M), ca 0.3 mi (0.5 km) NW junction of unnamed rd and Maple Creek Rd, 6.5 mi (10.5 km) SE Korbel, Humboldt So, CA (40.8360231 N, 123.8907322 W).

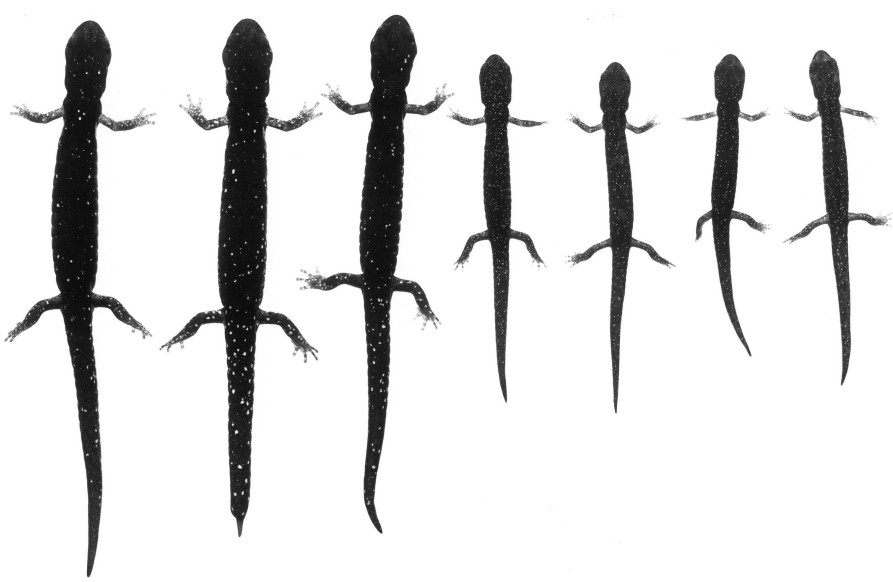

**Figure 7** **An ontogenetic series of seven *Aneides klamathensis* from ca. 4 km. NW Salyer, Trinity Co., CA.** The black and white photograph of living, anaesthetized specimens shows the gradual increase in size and number of whitish pigment cells as body size increases, and the relatively sparse number of such cells in this taxon (specimens arranged by JF Lynch and later preserved in MVZ; photo: Alfred Blaker, UC Berkeley Scientific Photographic Laboratory).

*Diagnosis:* A large (males to more than 80 mm SL; females to more than 85 mm SL) member of the *Aneides flavipunctatus* complex, subgenus *Aneides* (two subgenera in the genus, proposed by *Dubois & Raffaelli, 2012*, and discussed by *Wake, 2016*), distinguished from members of the subgenus *Castaneides* by larger size (*A. aeneus* less than 70 SL) and more robust body and tail, with relatively much shorter limbs and digits and blackish rather than greenish coloration. Distinguished from other members of subgenus *Aneides* as follows: from *A. hardii* by its much larger size (*A. hardii* less than 60 SL), more robust head, body and tail, and subdued sexual dimorphism; from the somewhat larger *A. lugubris* (some individuals exceed 100 mm SL) by darker ground coloration, more robust and less prehensile and tapered tail, and much shorter limbs and digits; from *A. ferreus* and *A. vagrans* by larger size (these species rarely exceed 75 mm SL), more robust and less prehensile and tapered tail, and much shorter limbs and digits. The new species is distinguished from other members of the *Aneides flavipunctatus* complex as follows: from *A. flavipunctatus* by geographic range and DNA sequences, from *A. iecanus* by having only relatively few small dorsal iridophores and in averaging 17 rather than 16 trunk vertebrae, from *A. niger* by coloration (*A. niger* is typically solid black with no whitish or gray markings).

*Description:* *Aneides klamathensis* is a large, robust plethodontid salamander that resembles other members of the *Aneides flavipunctatus* complex in its morphology. Standard length in the type series is 68.2–80.6, mean 74.5 +/− 3.4, for 11 adult males; 65.4–84.8, mean 71.7 +/− 5.3, for 10 adult females. Heads are very large and the jaw muscles of adults of both

sexes, but especially large males, are greatly expanded and bulge from the general outlines of the head. Head width of 11 males is 10.5–13.9, mean 12.5 +/− 0.92 (0.15–0.19, mean 0.17 +/− 0.01 times SL); 10 females 9.5–11.7, mean 10.5 +/− 0.65 (0.13–0.16, mean 0.15 +/− 0.01 times SL). The jaws are especially strong and bear a few large maxillary (1–7 each side) and mandibular (4–7 each side) teeth. Limbs are of modest length but relatively robust. Combined limb length (CLL) is 0.45–0.49, mean 0.47 +/− 0.02 SL in 11 males; CLL/SL 0.42–0.48, mean 0.46 +/− 0.02 in 10 females. Limb interval is 0.5–3, mean 1.9 +/− 0.84 in 11 males; 2–4, mean 2.1 +/− 0.24 in 10 females. Tails generally are robust but nearly all of them show signs of some regeneration, especially near the tip. Tails never are more than 0.82 SL. Digits, while long and slender, are less-so than those of other *Aneides* in California. Digits are modestly expanded terminally; the longest digits (#3) on the pes do not exceed 4.3.

***Description of the Holotype:*** The holotype is a large (76.7 SL) male that in preservative is generally very dark black over all its dorsal surfaces in preservation. There are scattered small whitish spots on the neck and sparsely along the lateral margins of the dorsum, with some scattered on the tail and its lateral surfaces. Whitish spots and blotches are prominent along the flanks of the trunk. Small but prominent whitish spots are present on both proximal and distal portions of the limbs and the dorsal surfaces of the hands and feet, although the fingers and toes are nearly unpigmented and appear gray in preservation. The tail is robust proximally but then sharpens to a point; the last 15% of the dorsal surface of the tail is unpigmented and appears gray. The tail is relatively short and is at least partially regenerated near the tip. The snout is broad and relatively flat, and it is broadly rounded at its tip. Nasolabial protuberances are modest in development and are unpigmented along the nasolabial groove. The eyes are moderately large and prominent but not very protruded from the rest of the head. The enlarged jaw muscles bulge outward well beyond the eyes. Integumentary grooves of the head are relatively prominent, and they tend to lose pigment in their deepest extents. The neck region is well defined. The limbs are relatively short, as are the digits. The first digits of both manus and pes are small and not prominent. The fourth digit of the manus and fifth of the pes are much shorter than the preceding digit and they are about the same length, or slightly shorter, than the second digit. Even the longest digits are only modestly expanded distally, with well developed subterminal pads. Ventral surfaces are generally dark black. However, there is a large and prominent ovoid mental gland that is lightly pigmented to unpigmented, and the pale area occupies about 40% of the gular region. The gular fold is unpigmented, as are the palms of the manus and pes. There is a general loss of pigment on the undersides of the limbs, giving them a generally gray appearance. The last 20% of the tail is unpigmented.

***Measurements (in mm), limb interval and tooth counts of the holotype:*** SL 78.7, Tail (regenerated) 50.7, Ax-Gr 42.8, S-G 15.7, HW 12.0, FLL 15.9, HLL 18.5, RH 6.3, RF 8.2, Minimal distance between eyes 5.5, Internarial distance 3.0, Horizontal eye diameter 3.4, Shoulder width 9.6, Length 3rd toe 3.7, Length 5th toe 2.4, Eye to nostril 3.4, Eye to snout 4.8, Width of mental gland 4.2, Length of mental gland 4.0, Head depth 6.1, Eyelid width 2.3, Eyelid length 3.6, Snout to forelimb 22.4, Diameter of external naris 0.4, Distance snout extends beyond mandible 1.2, Snout to anterior margin of vent 69.6, Tail width 5.6, Tail

depth 5.5, Maximal width of broadest toe 0.8, Limb interval 2.5, Number of costal grooves 15, Premaxillary teeth 11, Maxillary teeth 3-2 small anterior and 2-2 enlarged posterior, Vomerine teeth not countable.

***Coloration:*** Paedomorphic coloration in the sense of *Larson (1980)* and *Lynch (1981)*, consisting of heavy frosting of greenish gray pigment overlying black ground color on the dorsal surfaces and especially on the flanks of the trunk, where there is a sharp boundary with the generally black ventral coloration. Whitish to cream-colored or faint yellow spots of small to moderate size are evident on the dorsal surfaces of the limbs but are widely scattered and few in number on other dorsal surfaces (Figs. 6 and 7).

***Coloration of the Holotype in life*** (from field notes of SB Reilly): Intense whitish spotting/frosting on lateral sides and arms (Fig. 6A), light speckling on belly, and gold/gray frosting on back (Figs. 6B–6C and 7).

***Osteology:*** Information was derived from 7 adult males and 6 adult females, cleared and singly stained many years ago and only useful for some details. The largest individual is an 88.5 SL female from Hyampom, Trinity Co.; the smallest individual is 67.3 SL from Weitchpec, Humboldt County. One female has 18 trunk vertebrae and a female and a male have 16; all others have 17 trunk vertebrae. Vomerine teeth, not countable in the strongly jawed preserved specimens, are small and few in number. The smallest individual has no vomerine teeth. In other specimens, numbers range from 2–2 (the largest individual) to 5–5 (a 71.5 SL female). Premaxillary teeth vary from 4–8 in males and 3–7 in females. Large maxillary teeth range from 2–4 per bone in both sexes; small maxillary teeth range from 0 to 3 per bone in males and from 1 to 3 in females. Large mandibular teeth range from 2–4 in males and 1–4 in females.

The osteology of *Aneides flavipunctatus* was described in detail by *Wake (1963)* based on 13 singly cleared and stained and two skeletonized specimens. This was a sample of mixed origin, in part *Aneides flavipunctatus* (Mendocino Co.) and in part *Aneides niger* (Santa Cruz Co.). We studied the osteology of three individuals of *Aneides klamathensis* in detail and compared them to the descriptions in *Wake (1963)*. Two large female (MVZ 18468–88.5 SL, MVZ 199753–85.0 SL) and a large male (MVZ 184682–79.7 SL) were available as cleared and stained specimens in the MVZ collection. We compared them with the geographically most proximal member of *A. flavipunctatus,* a large male (MVZ 124079–80.1 SL) from Alderpoint, Humboldt Co, CA (40.1767267 N, 123.6102663 W), within two or three km of the nearest *A. klamathensis.* All specimens have 17 trunk vertebrae. MVZ 199735 and 124079 are especially well prepared individuals doubly stained for bone and cartilage; these were used for making detailed comparisons with the descriptions of *Wake (1963)*. We found no differences between the two species compared, nor with the published descriptions.

The skulls are very solid and well articulated and closely resemble the descriptions and figures (especially Figs. 3C & 3D) in *Wake (1963)*. Notable features are the tight articulation of the frontals and parietals on the skull roof, the well articulated facial region of the skull, and the prominent, high crests on the oticoccipitals. These crests extend far posterior and form the most caudal portion of the skull. There is no indication of any coossification of the skin of the snout and the underlying bones (as is prominent in *A. lugubris*, *Wake, 1963*).

 

The premaxillary and maxillaries are especially stout bones. The posterior part of each maxillary lacks teeth and is shaped like a cleaver, extending ventrally to the level of the tips of the enlarged maxillary teeth. The edentulous portion occupies more than half the length of the entire bone. Each maxillary has an interlocking articulation with the adjoining prefrontal. The ascending processes of the premaxillary envelop the internasal fontanelle, but while they approach each other behind the fontanelle, they do not articulate. The mandible is a large, robust bone and the jaw suspension, especially the quadrate and squamosal, are robust bones that have strengthening struts aligned more or less vertically. The bodies of the vomers are large and well articulated with each other along the midline. However, the posterior parts of the bones are reduced in size. Preorbital processes are absent and the teeth are relatively very small and few in number. Posterior vomerine patches of teeth approach each other at the midline but they are not in contact.

The jaw dentition is remarkably large and strong, but the teeth are not numerous. While the maxillary may bear as many as six teeth (a female), typically only two or three are both enlarged and ankylosed. Teeth are replaced from anterior to posterior along the jaw, with alternate positional replacement. Thus, two or even three rows of teeth are present and while only at most three are ankylosed at any one time, the larger replacement teeth protrude from the skin and have some degree of function. The same pattern is found on the dentary, but the ankylosed teeth are fewer in number and are even longer and stronger than those on the maxillaries. From only one to three ankylosed teeth are found in the four specimens studied in detail. The longest maxillary and dentary teeth are strongly recurved and cylindrical, not or only slightly flattened and very sharp; they appear formidable.

Premaxillary teeth are much smaller than those of the other jaw bones but still relatively large and well developed in comparison with most plethodontid salamanders. The single bone bears from six to eight ankylosed teeth.

In contrast the anterior vomerine teeth are very small and range in number from two to four. Posterior vomerine teeth are very small and arrayed in patches that add teeth laterally and shed them medially. At a given level midway through a patch there are from six to eight diagonal rows, each containing on the order of ten to twelve teeth.

All elements of the well-developed hyobranchial apparatus are cartilaginous except for the unpaired and unarticulated urohyal, which in this species is reduced to a tiny bit of bone on the ventral midline. The ceratobranchials and basibranchial are longer than the relatively short, tapering epbranchials, which are curved around the neck region, rising posterodorsally. The basibranchial has a pair of short radii and a piece that extends forward from their articulation to end in a small knob.

The forelimbs and hind limbs are as described by *Wake (1963)*. There are eight carpal elements and nine tarsal elements; distal tarsal 5 is larger than distal tarsal 4 and articulates with the intermedium. The phalangeal formulae are 1-2-3-2 and 1-2-3-3-2. The terminal phalanges of the longest digits are enlarged and expanded distally. The tip is strongly flattened and recurved and is hook-like on its outer margins (sometimes said to be *Y*-shaped). The entire face of the terminal portion is serrated or scalloped.

***Geographic Distribution:*** This is the northernmost member of the *A. flavipunctatus* complex. It ranges southward from the upper reaches of the Applegate river drainage in

Jackson Co., extreme southern Oregon and the southern bank of the Smith River in Del Norte Co., CA, south through Del Norte and Humboldt counties to the Van Duzen River and its tributaries, and east along the Klamath and Trinity rivers into Trinity and western Siskiyou counties, CA (Fig. 5). The species is distributed mainly at elevations below 500 m elevation but is known to occur as high as about 1,000 m near Hilt, Siskiyou Co., CA, at the extreme northeastern extent of it range. For more details on the contact zone between *A. klamathensis* and *A. flavipunctatus* in inland Humboldt County see discussion below.

*Etymology:* The Klamath Mountains, for which this species is named, is one of eleven geomorphic provinces in California. These mountains are a rare east to west oriented range in northwestern California and southwestern Oregon, with an elevation extending above 2,750 m. The Klamath River flows through the length of the range, and other important rivers include the Trinity and branches of the Rogue (especially Applegate and Illinois). The range harbors rich biodiversity and endemism and is home to the largest number of conifers on Earth (about thirty species including eighteen in a single square mile [2.6 km$^2$]).

*Remarks:* The first suggestion that *Aneides flavipunctatus* was a multispecies complex was presented in the unpublished doctoral dissertation of *Lowe (1950)*. Populations from the north coastal portion of the range of the complex south to the Longvale region of Mendocino County, CA, were recognized as a distinct subspecies and assigned a manuscript name. This distinction was based on the apparent preference for rock talus microhabitats, as well as the gray or greenish frosted coloration, which Lowe hypothesized was adapted for crypsis. This form was thought to range "in the outer Coast Ranges of northern California from the Klamath River of northern Humboldt County southward into Mendocino County in the Laytonville-Longvale area, and westward to the coastline" (Lowe. 1950 p. 3). The southern part of this range extends far to the south of the range of *A. klamathensis*, and the proposed type locality ("Squaw Creek, 1.3 miles north of Cummings, Mendocino County, California", *Lowe, 1950* p. 56; approximately 39.831582 N, 123.650227 W) is from what is now recognized as the northern segment (Central Core Clade 1, mitochondrial clade CM; *Reilly & Wake, 2015*, Fig. 3A) of the relatively wide-ranging *A. flavipunctatus* (*sensu stricto*), based on our current analysis.

The first functional explanation for the frosted coloration of the northwestern populations was paedomorphosis; proportions and coloration typical of juveniles are retained into adulthood (*Larson, 1980*). Larson postulated that many of the shape and proportional differences of *A. klamathensis* may not be adaptive, but rather a byproduct of selection for cryptic coloration and associated juvenile proportions. *Lynch (1981)* conducted a detailed color analysis and also concluded that these populations maintained the typical juvenile brassy pigmentation into adulthood. He noted that adults differed from juveniles in having more deeply embedded iridophores, which gave them a darker copper-toned color rather than the yellowish green coloration of juveniles. Based on this retention of juvenile coloration and external proportions, *Lynch (1981)* found that what is now considered *A. klamathensis* had the highest "paedomorphism index" level of all populations of the *flavipunctatus* complex.

Larson included two populations of *A. klamathensis* in his allozymic study, his populations 7 and 8. They were not especially distinctive in any way, but they were relatively most differentiated with respect to *A. iecanus* (Nei D 0.099–0.149) and especially *A. niger* (Nei D 0.170–0.215). The range of values of Nei D with respect to *A. flavipunctatus* was 0.040–0.117. The only fixed difference (for the allozyme *Got-1*) was in comparison to *A. iecanus.*

Population genetic analysis of samples from the region found that the Klamath watershed may have acted as a Pleistocene refugium, and that the Smith and Rogue River watersheds have been recently colonized from Klamath River populations (*Reilly et al., 2013*). While *A. klamathensis* has not been found in sympatry or close parapatry with *A. iecanus*, it is parapatric with *A. flavipunctatus* in southern Humboldt County south of the Van Duzen River (*Reilly & Wake, 2015*). By studying variation in an mtDNA gene (ND4) along a north-south transect in southeastern Humboldt County, we narrowed this zone of contact to ∼3 km, between Dobbyn Creek and the town of Blocksburg (Figs. 2 and 5). Coalescent analysis of 13 nuclear loci estimated that gene flow from *A. klamathensis sp. nov.* into *A. flavipunctatus* is $2Nm = 0.25$, with $2Nm = 0.53$ in the opposite direction (*Reilly & Wake, 2015*). This suggests that infrequent migration and hybridization occurs between these lineages. Both the mtDNA phylogeny and nDNA population clustering analyses find a narrow, unambiguous zone of contact (*Reilly & Wake, 2015*). The mtDNA + nDNA and nDNA coalescent species tree analyses find that *A. klamathensis* sp. nov. is sister to *A. flavipunctatus*, a finding that is in disagreement with the mtDNA-only phylogeny (which places *A. klamathensis* as sister to *A. iecanus,* Fig. 8).

*Dubois & Raffaelli (2012)* used the name *Aneides* ''sequoiensis'' and added the date ''1950''. This is a manuscript name from the unpublished portion of the doctoral thesis of *Lowe (1950)* and hence is unavailable (a *nomen nudum*). Furthermore, while it was intended to refer to the taxon we have named *Aneides klamathensis*, the type locality used by Lowe is within the geographic range of *Aneides flavipunctatus,* in the sense of the use of that taxon in this work.

### *Aneides niger* Myers & Maslin 1948

*Autodax iëcanus* (part)—Van Denburgh, 1896
*Aneides flavipunctatus* (part)—*Storer, 1925*
*Aneides flavipunctatus niger*—*Myers & Maslin, 1948*
Santa Cruz Black Salamander
Fig. 9

**Holotype**: Originally Stanford Natural History Museum (SNHM) #2938, currently CAS SUA 2918.
**Type Locality**: ''near the forks of Waddell Creek, Santa Cruz County'' (approximately 37.133876 N, 122.267535 W, 26 m elevation), CA; collected by GS Myers and MW Brown.
**Diagnosis**: A large (males and females exceed 80 mm SL) member of the *Aneides flavipunctatus* complex, subgenus *Aneides,* distinguished from members of the subgenus

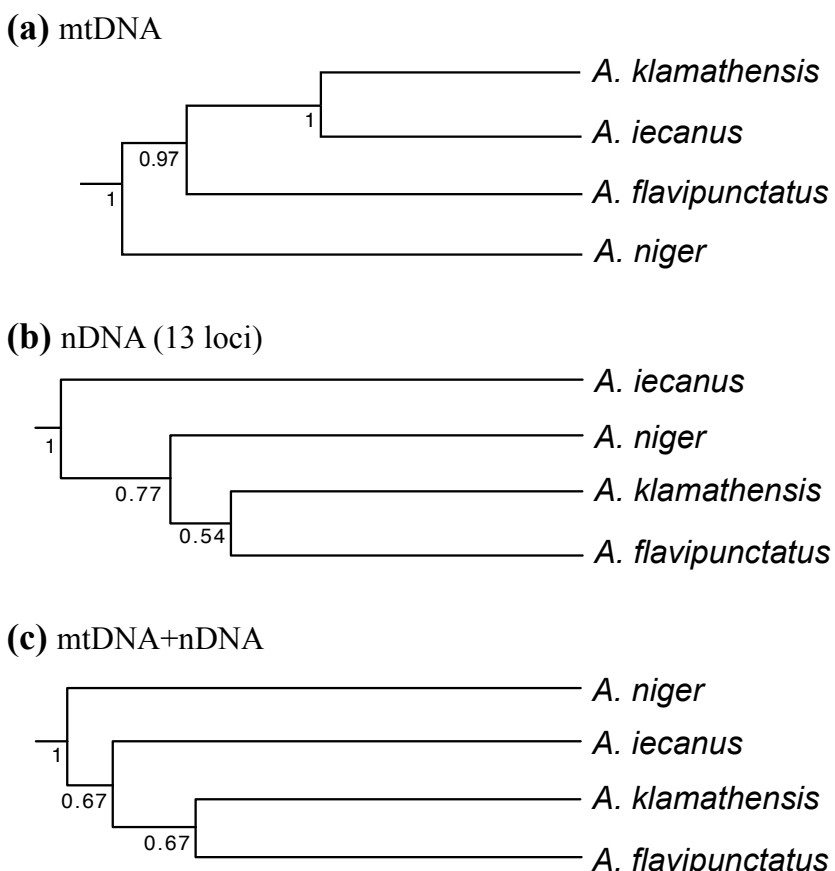

**(a)** mtDNA

**(b)** nDNA (13 loci)

**(c)** mtDNA+nDNA

**Figure 8 Phylogenetic relationships of the *Aneides flavipunctatus* complex.** (A) BEAST analysis of the *ND4, cytb,* and *12S* mitochondrial genes, (B) *BEAST analysis of 13 nuclear loci, and (C) *BEAST analysis of three mtDNA and 13 nDNA loci. Trees adapted from *Reilly & Wake (2015)*.

*Castaneides* by larger size (*A. aeneus* less than 70 SL), rounded rather than flattened head and body, and more robust body and tail, with relative much shorter limbs and digits and blackish rather than greenish coloration. Distinguished from other members of subgenus *Aneides* as follows: from *A. hardii* by its much larger size (*A. hardii* less than 60 SL), more robust head, body and tail, and subdued sexual dimorphism; from the somewhat larger *A. lugubris* (some individuals exceed 100 mm SL) by darker ground coloration, more robust and less prehensile and tapered tail, and much shorter limbs and digits; from *A. ferreus* and *A. vagrans* by larger size (these species rarely exceed 75 mm SL), more robust and less prehensile and tapered tail, and much shorter limbs and digits. This species is distinguished from other members of the *Aneides flavipunctatus* complex by its nearly uniform black coloration in adults; juveniles have numerous tiny white dorsal spots that are lost progressively at larger sizes (*Lynch, 1981*) (Fig. 9); it is further distinguished from *A. iecanus* by having an average number of trunk vertebrae of 17 rather than 16.

**Description:** *Aneides niger*, like other members of the *Aneides flavipunctatus* complex, is a large, robust salamander. *Lynch (1981)* reported SL for combined population samples from

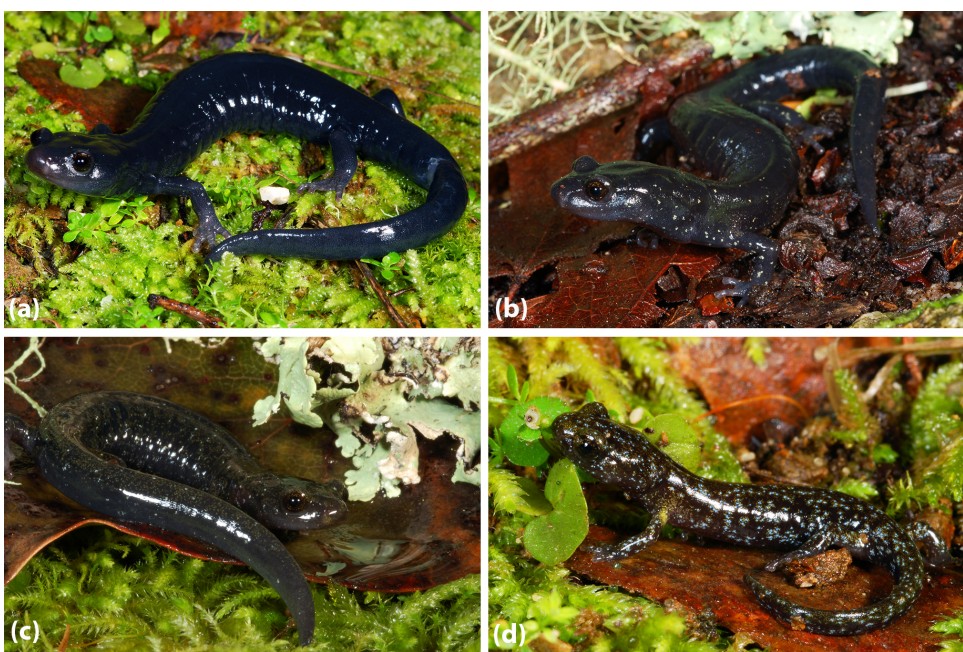

**Figure 9** *Aneides niger* **from the campus of UC Santa Cruz.** (A) An adult female exhibiting a pure black coloration, (B) an adult male with very small yellow flecks, (C) a juvenile which retains some xanthophore pigment frosting on the dorsal surface, and (D) a hatchling with bright bluish iridophore pigment flecks, xanthophore frosting and bright yellow color at the base of the limbs (common in all four species). All specimens released (photos: M Mulks).

Santa Clara (13 males, mean SL 69.2; 16 females, mean SL 68.6) and Santa Cruz counties (15 males, mean SL 65.7; 17 females, mean SL 60.8). We measured samples of large adults from across the range and obtained values of SL of 68.8–85.7, mean 75.9 +/− 6.0 for 10 males; 58.3–73.7, mean 67.7 +/− 5.2 for 10 females. As in other members of this complex, heads are large and laterally expanded behind the eyes in both sexes but especially so in large males. Head width of 10 males is 10.5–16.3, mean 12.9 +/− 1.8 (0.15–0.19, mean 0.17 +/− 0.01 times SL); 10 females 8.9–10.9, mean 10.2 +/− 1.8 (0.14–0.17, mean 0.15 +/− 0.01 times SL). The strong jaws bear few maxillary and mandibular teeth, but the longest one to three teeth are very large. Limbs are moderately long and robust. Combined limb length (CLL)/SL is 0.41–0.46, mean 0.43 +/− 0.02 in males; 0.42–0.49, mean 0.43 +/− 0.03 in females. Limb interval is 2.5–4, mean 3.2 +/− 0.54 in males; 2–3.5, mean 3.0 +/− 0.47 in females. Tails are robust and moderately long, but all show signs of regeneration. Only three of twenty individuals have tails that are 0.8 times SL or longer (the longest is 0.87 SL in the largest male). Digits are long and slender and are terminally expanded. The longest digit on the pes is 3.9.

*Geographic Distribution*: *Aneides niger* occurs only in the Santa Cruz Mountains on the lower San Francisco Peninsula in Santa Cruz, western Santa Clara, and extreme southern and eastern San Mateo counties, California (Fig. 5). The southernmost locality is at approximately 37°N. The species occurs from near sea level to elevations of approximately 800 m. This is the most semiaquatic member of the complex and individuals are commonly

encountered in the margins of rapidly flowing streams and in wet, rocky seeps. Individuals are rarely encountered far from water.

*Etymology*: The name refers to the solid black coloration of adults of this species.

*Remarks*: *Myers & Maslin (1948)* based their description of *A. f. niger* on the absence of spots (a similar coloration is found in coastal Sonoma and Mendocino Counties) and the disjunct distribution. *Lowe (1950)* concurred with this subspecific designation and added ecological evidence, specifically the semiaquatic microhabitat preference (Myers and Maslin had earlier noted its "hydrophilous" nature and compared its habitat to that of *Desmognathus fuscus*). However, *Lynch (1974)* suggested that the coloration and distribution criteria used by Myers and Maslin were largely invalid because they had not obtained sufficient specimens from localities throughout the range of the complex to be able to conduct appropriate comparisons. Later, *Lynch (1981)* concluded that Myers and Maslin's claim that *A. f. niger* had relatively shorter limbs was driven by a smaller mean SVL of *A. f. niger* specimens (65 mm) examined compared to specimens examined from the main range (75 mm), and that this difference was ontogenetic in nature.

*Lynch (1981)* meticulously documented coloration characters from throughout the range and found that iridophore size in *A. f. niger* did not increase with body size, as in populations from the main part of the geographic range of the complex. Furthermore, *A. f. niger* showed the most dramatic reduction of dorsal iridophores and a denser melanophore network on the chin of any population. His morphological analysis revealed that *A. f. niger* has slightly shorter tails relative to body length than main range populations, and that 95% of *A. f. niger* samples examined contained 17 trunk vertebrae while the nearest populations in Sonoma and Napa Counties had a modal count of 16 trunk vertebrae. Lynch created a paedomorphism index to quantify the degree to which each population of *A. flavipunctatus* retained juvenile characteristics. He found that *A. f. niger* contained the lowest paedomorphism index score of any *A. flavipunctatus* populations, suggesting that they retain fewer juvenile characteristics than any other population.

Both Lowe and Lynch noticed that *A. f. niger* was active at the surface by day in atmospheric conditions of nearly 100% humidity in the wet, heavily shaded streamside habitat they frequent. Individuals often are found in contact with, or submerged in, standing water. The only other population with a similar microhabitat preference is the Shasta County population (*A. iecanus*), which may explain the phenetic similarity of *A. niger* to *A. iecanus* calculated by Lynch, possibly a manifestation of convergent evolutionary adaptations of these two species to a semi-aquatic way of life. However, a PCA of environmental space revealed that *A. f. niger* occupies a distinct environmental space when compared to the rest of the range (*Rissler & Apodaca, 2007*).

*Highton (2000)* reanalyzed a large number of allozymic studies of salamanders and offered alternative interpretations of their taxonomic significance. One of these was the *Aneides* study of *Larson (1980)*, to which he devoted a short paragraph. Highton advocates a level of allozymic genetic distance of about 0.15 as approximating the level appropriate for species recognition. He found three such groups in Larson's *Aneides flavipunctatus* data and concluded "the taxonomic hypothesis that these groups are three different species is more strongly supported by the available evidence than the hypothesis that they represent a single

species". The allozymic study of *Larson (1980)* obtained estimates of Nei genetic distance from *A. niger* as follows: *A. klamathensis* 0.170–0.215, *A. iecanus* 0.222, *A. flavipunctatus* 0.130–0.182. Additionally, Larson estimated that *A. f. niger* had first diverged from *A. iecanus* nearly 3 million years ago and had subsequently diverged from the main range approximately 2.4 million years ago.

Evidence has only grown stronger in the intervening decades, including data from mtDNA and nuclear gene sequences (reviewed by *Reilly & Wake, 2015*) and the new morphometrical analysis presented herein. While *Rissler & Apodaca (2007)* suggested that *A. f. niger* represented a distinct species they did not formally recognize it as such. Nevertheless, the on-line database Amphibian Species of the World 6.0, an Online Reference (http://research.amnh.org/vz/herpetology/amphibia/Amphibia/Caudata/Plethodontidae/Plethodontinae/Aneides/Aneides-niger) recognized the taxon as a full species, citing Rissler and Apodaca "by implication" (also citing, *Collins & Taggart, 2009*; *Dubois & Raffaelli, 2012*, although both are simply lists).

## Redescription of *Aneides iecanus* (Cope 1883)

*Plethodon iëcanus*—*Cope, 1883*
*Anaides iëcanus*—*Cope, 1886*
*Autodax iëcanus*—*Cope, 1889*
*Aneides iecanus*—*Grinnell & Camp, 1917*
*Aneides flavipunctatus* (part)—*Storer, 1925*
Shasta Black Salamander
Figs. 10–11

*Holotype*: ANSP 14061 (*Fowler & Dunn, 1917*)
*Type Locality*: "—near the United States fish-hatching establishment on the McCloud River, in Shiasta (*sic*) County". Collected by ED Cope.
*Diagnosis*: A large (some individuals exceed 80 mm SL) member of the *Aneides flavipunctatus* complex, subgenus *Aneides,* distinguished from members of the subgenus *Castaneides* by larger size (*A. aeneus* less than 70 SL), rounded rather than flattened head and body, more robust body and tail, with relative much shorter limbs and digits and blackish rather than greenish coloration. Distinguished from other members of subgenus *Aneides* as follows: from *A. hardii* by its much larger size (*A. hardii* less than 60 SL), more robust head, body and tail, and subdued sexual dimorphism; from the somewhat larger *A. lugubris* (some individuals exceed 100 mm SL) by darker ground coloration, more robust and less prehensile and tapered tail, and much shorter limbs and digits; from *A. ferreus* and *A. vagrans* by larger size (these species rarely exceed 75 mm SL), more robust and less prehensile and tapered tail, and much shorter limbs and digits. This species is distinguished from other members of the *Aneides flavipunctatus* complex by its heavily speckled head, body and tail (*Lynch, 1981*) (Figs. 10 and 11); it is further distinguished from both *A. klamathensis* and *A. niger* by having an average of 16 rather than 17 trunk vertebrae.

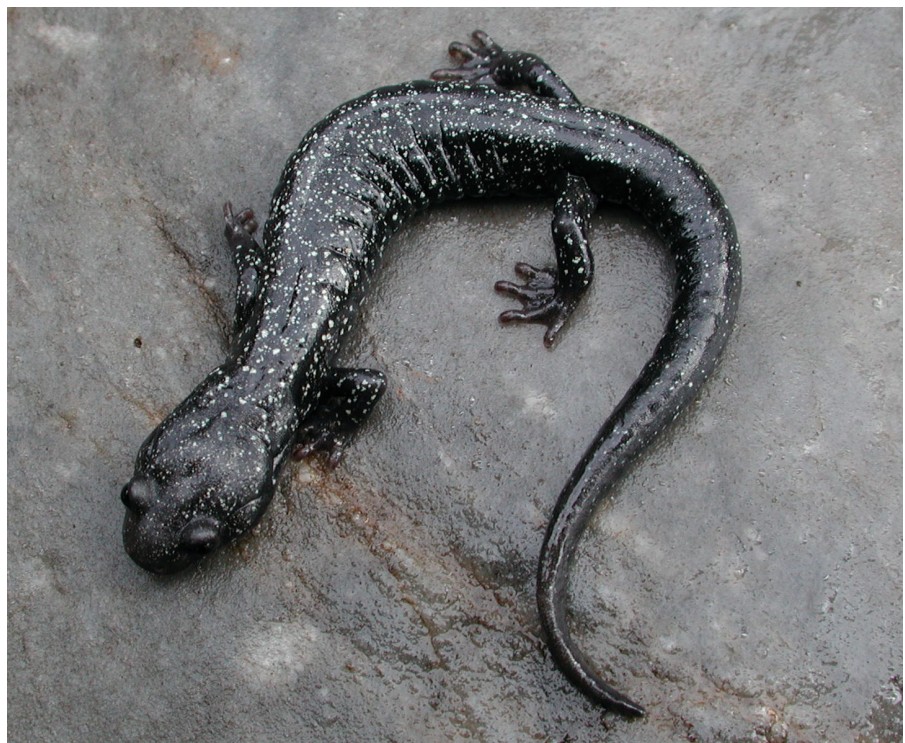

**Figure 10 Adult *Aneides iecanus*.** Photographed in life near Dekkas Rock, east side of McCloud Arm, Lake Shasta, Shasta Co., CA (40.871418 N, 122.223491 W). Note dense scattering of moderately small whitish pigment cells. Specimen released (Photo: DB Wake).

***Description*:** *Aneides iecanus*, like other members of the *Aneides flavipunctatus* complex, is a large, robust salamander. We measured samples of large adults from across the range and obtained values of SL of 69.9–81.6, mean 75.8 +/− 3.9 for 10 males; 60.9–78.7, mean 70.1 +/− 5.5 for 10 females. As in other members of this complex, heads are large and laterally expanded behind the eyes in both sexes but especially so in large males. Head width of 10 males is 11.0–13.3, mean 12.5 +/− 0.65 (mean 0.16 +/− 0.01 times SL); 10 females 8.9–10.8, mean 10.0 +/− 0.64 (mean 0.14 +/− 0.01 times SL). The strong jaws bear few maxillary and mandibular teeth, but the longest one to three teeth are very large. The large teeth are cylindrical rather than flattened. Limbs are moderately long and robust. Combined limb length (CLL) is 26.5–35.9, mean 30.0 +/− 3.4 in males (CLL/SL 0.43–0.50, mean 0.41 +/− 0.13); CLL 27.0–33.0, mean 30.6 +/− 1.7 in females (CLL/SL 0.41–0.47, mean 0.44 +/− 0.02). Limb interval is 2–3, mean 2.2 +/− 0.35 in males; 2.5–4, mean 3.4 +/− 0.52 in females. Tails are robust and moderately long but most show signs of regeneration. The longest tails reach 0.85 SL in males and 0.78 SL in females. Digits are long and slender, and are terminally expanded. The longest digit on the pes is 3.8 in a female.

***Geographic Distribution*:** The species is known from north central and western Shasta County, California, as well as extreme southeastern Siskiyou County in the vicinity of Castle Crags, California, but the identity of scattered specimens from the western margin

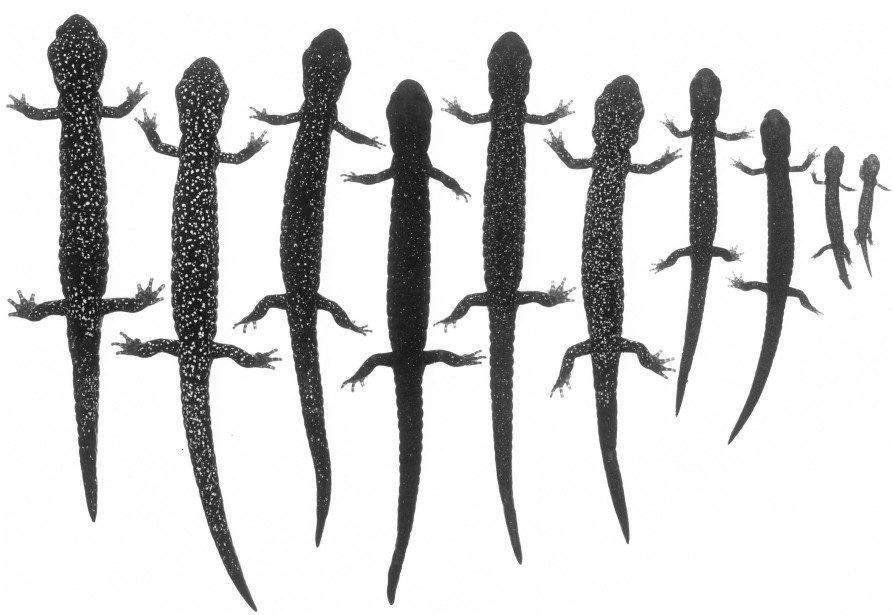

**Figure 11** **An ontogenetic series of ten *Aneides iecanus* from Castle Crags region, Shasta Co., CA.** The black and white photograph of living, anaesthetized specimens shows the gradual (but not monotonic) increase in size and number of whitish pigment cells with size, and the relatively dense number of such cells in adults of this taxon. Note the greatly enlarged jaw muscles of adults (specimens arranged by JF Lynch and later preserved in MVZ; photo: Alfred Blaker, UC Berkeley Scientific Photographic Laboratory).

of the Sacramento Valley to the south has not been determined. *Aneides iecanus* occurs at elevations ranging between about 300 m (near the surface of Lake Shasta) to over 1,000 m (in the Castle Crags area). Populations along the inner margins of the Coast Ranges in western Tehama and Glenn Counties (see map Fig. 5) may be assignable to *Aneides iecanus*, but further surveys including morphological and genetic analyses are needed.

***Etymology***: The species name was derived from the local Native American word "Iëka", which according to Cope refers to Mount Shasta.

***Remarks***: The type locality, near the old federal fish hatchery on the McCloud River, named Baird, now lies under the waters of the reservoir, Shasta Lake. The species is widespread in the Shasta Lake area, and extends as far north as the Castle Crags area right on the Siskiyou-Shasta county border. Cope either was unaware of the description of *P. flavipunctatus* three years earlier, or of its relatedness to *P. iecanus* (*Cope, 1889*, placed the two taxa in different genera). The taxon was synonymized with *A. flavipunctatus* by *Storer (1925)*.

*Larson (1980)* concluded that populations of *A. flavipunctatus* from Shasta County were divergent within the *flavipunctatus* complex. The range of Nei D to other part of the complex are: to *klamathensis* 0.099–0.149, to *flavipunctatus* 0.132–0.209, and to *niger* 0.222. Larson estimated that the Shasta population had first diverged from the rest of the complex nearly 2.6 million years ago. *Rissler & Apodaca (2007)* estimated a mtDNA phylogeny that found Shasta County samples to be monophyletic, and sister to samples from the Klamath

Mountains. This finding was supported by subsequent mtDNA phylogenies estimated by *Reilly et al. (2013)* and *Reilly & Wake (2015)*, which suggested a mtDNA divergence of 4–6.8% corresponding to a divergence time in the mid-Pleistocene. Population clustering analysis of sequence data from 13 nuclear loci also found the Shasta populations to be distinct, with unique derived mutations at nearly every locus (*Reilly et al., 2013*). Coalescent analyses of gene flow between Shasta County and the Klamath Mountains populations found 2*Nm* values to be less than 1 in both directions, suggesting genetic isolation across the Trinity Mountains ridge separating Shasta and Trinity Counties (*Reilly et al., 2013*). When a species tree methodology is applied to the mtDNA + nDNA data, populations from Shasta County are recovered as sister to all main range populations (*A. flavipunctatus + A. klamathensis*). However, a species tree constructed from only nuclear loci is in agreement with Larson's finding that Shasta County populations constitute the basal lineage within the complex (*Reilly & Wake, 2015*) (see Fig. 8).

The genetic findings outlined above are reinforced by ecological and morphological findings from previous studies that show sharp breaks in microhabitat use, vertebral number, and coloration between Shasta salamanders and the nearest populations within the Klamath watershed to the west. *Lowe (1950)* considered the Shasta population to be a distinct sub-species, because they differed from Klamath populations in habitat use and color pattern. Subsequently, *Lynch (1974)* found that salamanders in Shasta County prefer shaded, streamside habitat, while salamanders in the Klamath watershed prefer exposed, rock talus habitat. Osteologically, Shasta Co. black salamanders average 16 vertebrae while Klamath River watershed black salamanders average 17 vertebrae (*Lynch, 1981*). With regards to coloration, Shasta Co. black salamanders have a black coloration with numerous small white spots (Fig. 11), while Klamath watershed black salamanders generally retain the juvenile xanthophore pigments into adulthood, which gives them a green, gold, or greyish frosted coloration (*Lynch, 1981*) (see Figs. 6 and 7).

## Redescription of *Aneides flavipunctatus* (Strauch 1870)

*Plethodon flavipunctatus*—*Strauch, 1870*
*Aneides flavipunctatus* (part)—*Storer, 1925*
Speckled Black Salamander
Figs. 12–15

**Lectotype**: "ZISP 156, "Californien (Neu-Albion)" "Leg: I. G. Wosnessensky, 1843" (Lectotype designated by *Milto & Barabanov, 2011*, p. 139; Fig. 14) (original syntypes ZISP 155 [now lost], ZISP 157). *Milto & Barabanov (2011)* interpreted "Neu-Albion" as [Albion, Mendocino County, California, USA].

**Diagnosis**: A large (some individuals exceed 80 mm SL) member of the *Aneides flavipunctatus* complex, subgenus *Aneides,* distinguished from members of the subgenus *Castaneides* by larger size (*A. aeneus* less than 70 SL), rounded rather than flattened head and body, more robust body and tail, with relative much shorter limbs and digits and

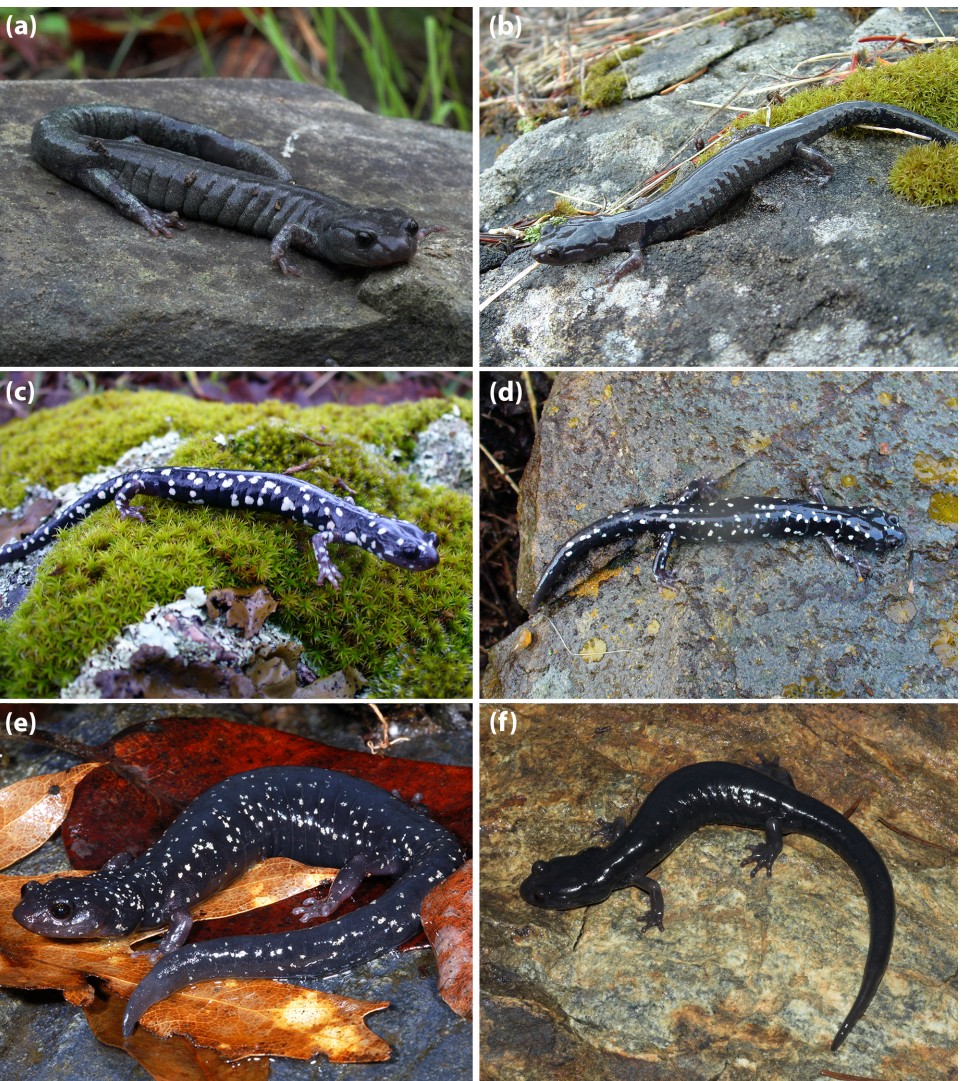

**Figure 12  A series of *A. flavipunctatus* exhibiting some of the more common color patterns.** (A) MVZ 264011 from ∼3 km W Miranda, Humboldt Co., CA, (B) MVZ 269402 from ∼6 km SE Scotia (locality # 14 from Fig. 2), Humboldt Co., CA, (C) MVZ 264056 from ∼7 km S Potter Valley, Mendocino Co., CA, (D) MVZ 269459 from ∼7 km E Geyserville, Sonoma Co., CA, (E) a female (found and released) from ∼10 km W Geyserville, Sonoma Co., CA, and (F) MVZ 264023 from ∼5 km E Fort Bragg, Mendocino Co., CA. (Photos: (A) A Gottscho; (B, C, D, F) S Reilly; (E) M Mulks).

blackish rather than greenish coloration. Distinguished from other members of subgenus *Aneides* as follows: from *A. hardii* by its much larger size (*A. hardii* less than 60 SL), more robust head, body and tail, and subdued sexual dimorphism; from the somewhat larger *A. lugubris* (some individuals exceed 100 mm SL) by darker ground coloration, more robust and less prehensile and tapered tail, and much shorter limbs and digits; from *A. ferreus* and *A. vagrans* by larger size (these species rarely exceed 75 mm SL), more robust and less prehensile and tapered tail, and much shorter limbs and digits. This species is distinguished from other members of the *Aneides flavipunctatus* as follows: from *A. klamathensis* in

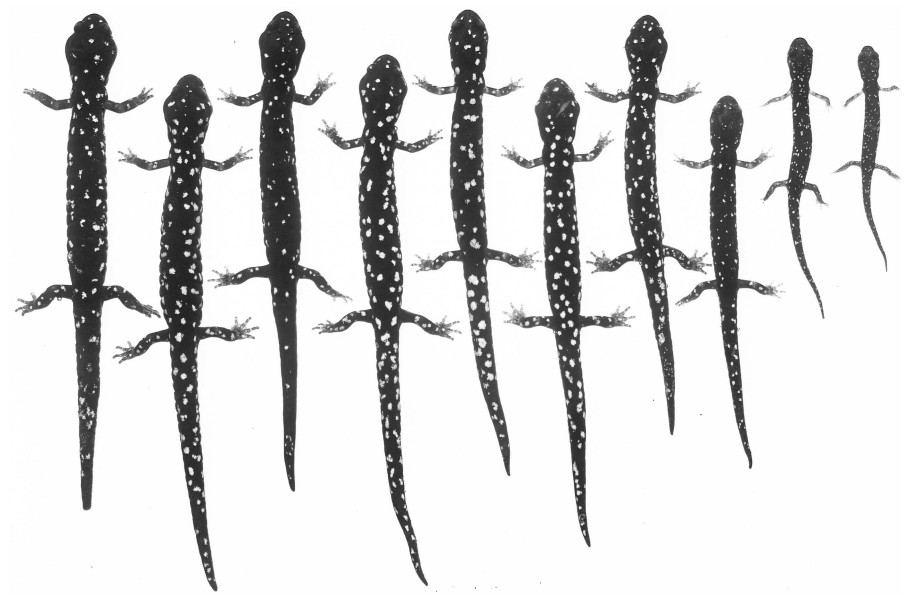

**Figure 13** An ontogenetic series of ten *Aneides flavipunctatus* from area about 3.2 km E, 0.5 km S Geyserville, Sonoma Co., CA. The black and white photograph of living, anaesthetized specimens shows the relatively numerous and large whitish pigment cells which increase with size, and the relatively dense concentration of such cells in adults of southern inland populations of this taxon (specimens arranged by JF Lynch and later preserved in MVZ; photo: Alfred Blaker, UC Berkeley Scientific Photographic Laboratory).

having variable coloration but generally lacking the frosted dorsal coloration, especially in the southern parts of its range, and only a few populations have 17 trunk vertebrae; from *A. iecanus* in it more variable coloration and habitat preferences; from *A. niger* in its more variable coloration but rarely with so few iridophores.

***Description:*** *Aneides flavipunctatus* is a large, robust salamander. *Reilly & Wake* (*2015*, Fig. 3A) recognized two major molecular-based clades, northern (Central Core, Clade 1) and a southern (Central Core, Clade 2) groups of populations. *Lynch (1981)* presented information on mean SL and possible sexual dimorphism for relatively large samples of full adults from populations representing each clade, as follows: Clade 1: Lynch population 001 (Salmon Point), M 65, F 68; Population 007 (McGuire Hill), M 65.4, F 69 (statistically significant difference); Population 158 (Leggett), M 74.7, F 69.2; Population 161 (Usal) M 70.9, F 68.8; Population 187 (Alderpoint), M 72.8, F 70.9. Clade 2: Population 010 (Navarro), M 69.3, F 66.7; Population 055 (Skaggs Springs) M 64.4, F 63.7; Population 063 & 070 (Guerneville) M 68.5, F 68.1; Population 111 (Potter Valley), M 64.1; F 63.6; Population 121 (Geyserville), M 71.1, F 64.5 (statistically significant difference). We measured samples of large adults selected from populations in the heart of the respective ranges and obtained values of SL, as follows. Clade 1: 64.4–80.3. mean 72.8 +/− 5.0 for 10 males; 67.2–78.1, mean 71.2 +/− 3.2 for 11 females. Clade 2: 64.0–71.7, mean 68.0 +/− 2.6 for 10 males; 63.3–69.9, mean 66.6 +/− 2.1 for 10 females. As in other members of this complex, heads are large and laterally expanded behind the eyes in both sexes but

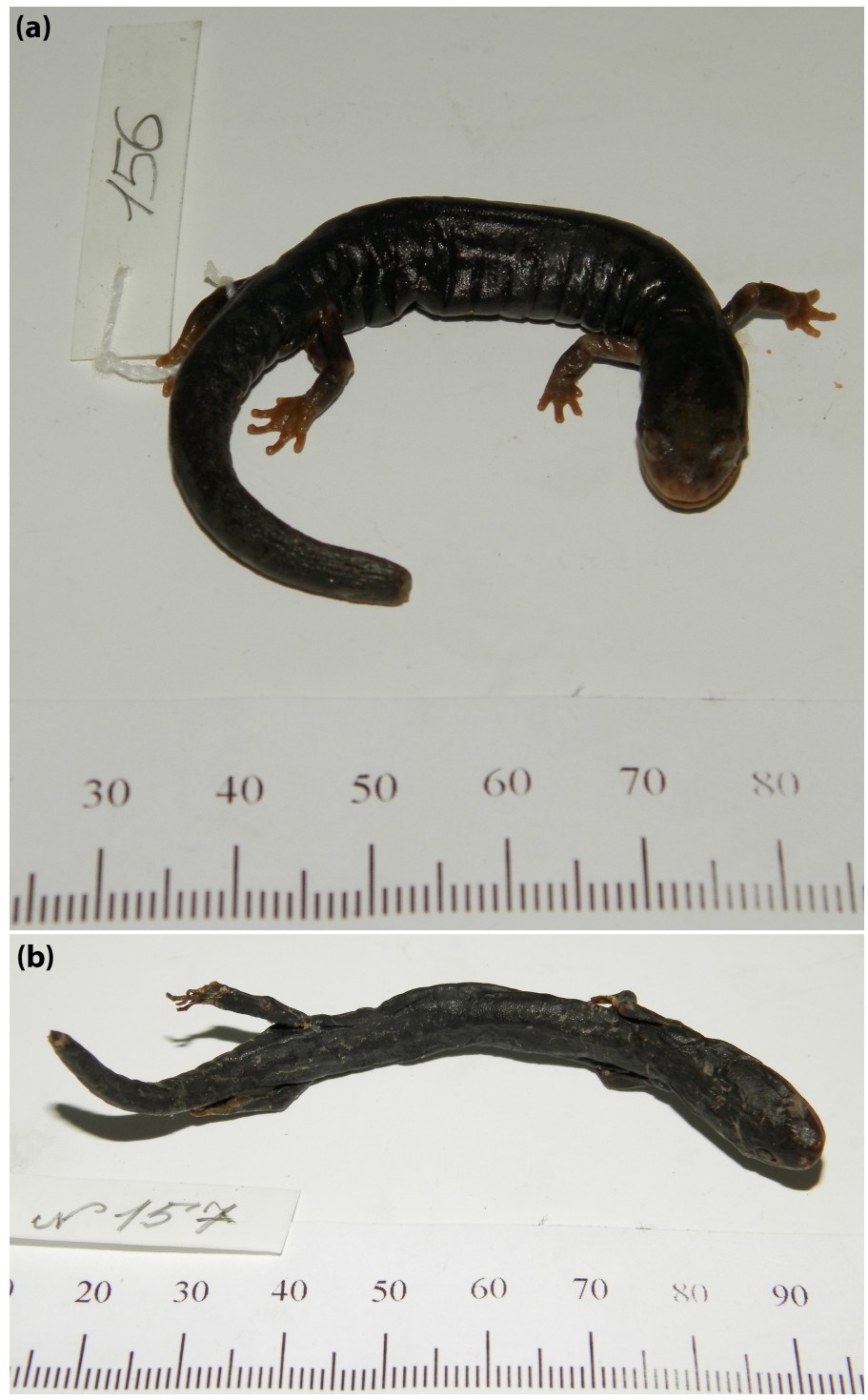

**Figure 14  Lectotype and paralectotype of *A. flavipunctatus*.** (A) Photo of the lectotype of *Plethodon flavipunctatus* Strauch 1870, ZISP (Zoological Institute St. Petersburg) 156, courtesy of N Ananyeva and K Milto, Zoological Collections, Academy of Science, St. Petersburg, Russia. (B) Photo of the sole remaining paralectotype of *Plethodon flavipunctatus* Strauch 1870, ZISP 157, courtesy of N. Ananyeva and K. Milto, Zoological Collections, Academy of Science, St. Petersburg, Russia.

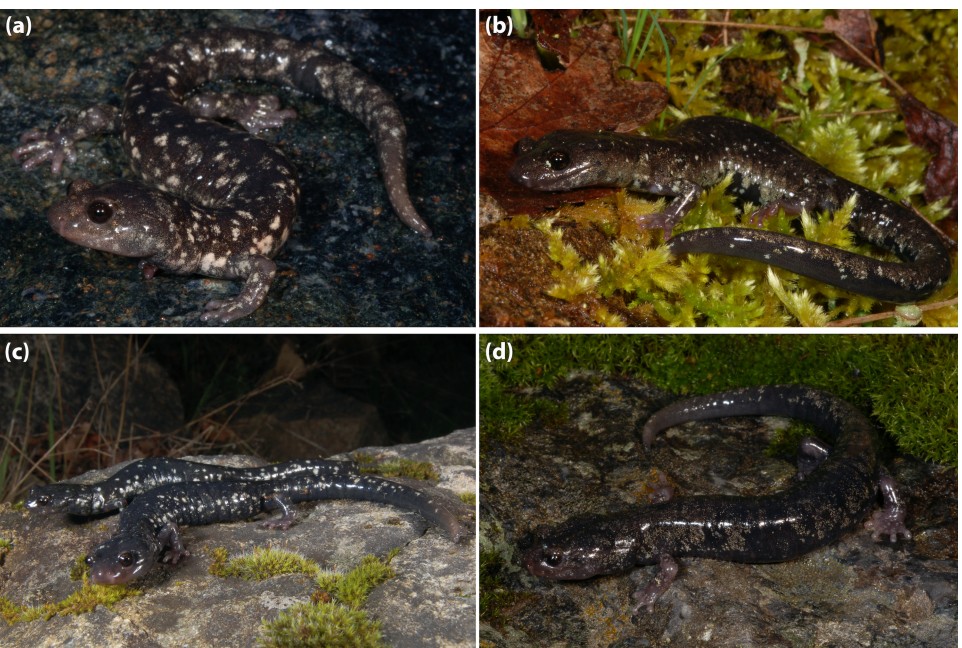

**Figure 15  A series of *A. flavipunctatus* from the Longvale/Laytonville region of Mendocino Co., CA.** This area is identified as a contact zone between two genetically differentiated groups of populations (see *Reilly, Marks & Jennings, 2012*; *Reilly & Wake, 2015*). (A) MVZ 264029 from 7 km S Laytonville, (B) MVZ 264032 from 16 km N Laytonville, (C) male (front) MVZ 264028 and female (behind) MVZ 264049 from Longvale, and (D) MVZ 264031 from 9 km N Laytonville. (photos: M Mulks).

especially so in large males. Head width for the Clade 1 sample of 10 males is 10.4–15.2, mean 11.9 +/− 1.4 (0.13–0.16, mean 0.15 +/− 0.01 times SL); 11 females 8.9–11.8 mean 10.8 +/− 0.98 (0.13–0.16, mean 0.15 +/− 0.01 times SL); for Clade 2 sample of 10 males is 10.3–11.7, mean 11.1 +/− 2.6 (0.15–0.18, mean 0.16 +/− 0.03 times SL); 10 females 92–11.1, mean 10.0 +/− 0.65 (0.14–0.16, mean 0.15 +/− 0.01 times SL). The strong jaws bear few maxillary and mandibular teeth, but the longest one to three teeth are very large. Limbs are moderately long and robust. Combined limb length (CLL) for the Clade 1 sample of 10 males is 32.3–36.8, mean 34.9; CLL/SL 0.45–0.49, mean 0.47 + 0.02; 11 females 31.0–35.5, mean 33.0 +/− 1.5; CLL/SL 0.42–0.48, mean 0.46 +/− 0.02; for Clade 2 sample of 10 males 29.1–31.9, mean 30.2 +/− 0.41; for 10 females is 27.4–31.4, mean 29.3 +/− 1.3. Limb interval for Clade 1 is 1–3, mean 2.6 +/− 0.6 for 10 males; 2- 4, mean 3.2 +/− 0.3 for 11 females; for Clade 2 2–3, mean 2.25 +/− 0.95 in 10 males; 2.5–4, mean 3 +/− 0.4 in 10 females. Tails are robust and moderately long but most show signs of regeneration. The longest tails reach 0.94 SL in males and 0.87 SL in females. Digits are long and slender, and are terminally expanded. The longest digit on the pes for Clade 1 males is 3.5 and 4.5 in a female; for Clade 2 males 3.2 and 4.9 in a female (females generally have longer digits).

*Geographic distribution*: From northern Sonoma and Napa counties north into southern Humboldt County near Cape Mendocino and Larabee Creek, east to the interior edge of the coast ranges. As mentioned above, populations along the inner margins of the Coast

Ranges in western Tehama and Glenn Counties (see map Fig. 5) are of unknown status, and further surveys including morphological and genetic analyses of these populations are needed to confirm their taxonomic designation.

**Etymology:** Strauch (1870) offered no explanation for his name, but presumably he assumed that the light spots then still evident in the types were yellowish, rather than whitish or cream-colored (the true color) in life.

**Remarks:** In previous comparison sections we have summarized the results of Larson's (1980) allozyme study and presented Nei genetic distances to *Aneides flavipunctatus*. The greatest value of Nei D is 0.209 to *A. iecanus*; all other values are 0.182 or less. Among population levels within *A. flavipunctatus* range from 0.023 to 0.117. These distances are not as great as one might expect based on the high degree of divergence within *A. flavipunctatus* in DNA (Reilly & Wake, 2015). The allopatry of *A. flavipunctatus, A. niger* and *A. iecanus* assures that the three taxa are readily distinguished, but it is difficult if not impossible to distinguish living *A. flavipunctatus* and *A. klamathensis* in the inland portions of their contact zone. The very detailed study of coloration and morphology by Lynch (1981) documents the great variation within *A. flavipunctatus* (see Figs. 1 and 12) while at the same time making clear that while *A. klamathensis* is distinguishable from most *A. flavipunctatus* by general color pattern and by details of coloration, where the ranges of the two approximate each other it can be difficult to separate the two. A genomic approach is likely needed to fully understand the dynamics of the southern Humboldt contact zone.

A detailed itinerary of the collector of the types, I. G. Voznesenskii, is available in Alekseev (1987). We go into some detail in following Voznesenskii's travels because it is important to try to identify the type locality, given the very extensive genetic substructuring of clade 2 of our revised *Aneides flavipunctatus*. He arrived in present-day Bodega Bay, about 18 miles (29 km) south of the Fort Ross settlement, on July 20, 1840, and slowly made his way north, collecting along the way. He visited the Russian Chernykh and Kostromitinov ranches, south of the "Slavianka" River (present-day Russian River), and reached Fort Ross in mid-August, 1840. The fort was then in its last days, a decision having been made in 1839 to abandon it (the land and much equipment was sold to the American John Sutter). Voznesenskii did not spend much time at the fort, but took excursions northward (as far as Cape Mendocino) and inland. He packed 13 crates and two kegs with materials to be returned to Russia and shipped them from San Francisco in October, 1840. Apparently, the salamanders in question were not a part of that collection, because they did not reach the Museum in St. Petersburg until 1843 (Milto & Barabanov, 2011). He spent a lot of time in late 1840 and the spring of 1841 in and around San Francisco Bay. From mid-April through May and June, 1841, he traveled the length of the Russian River and surrounding area, and on June 16th he became the first westerner to climb Mt. St. Helena (which he named, for the wife of the manager of Fort Ross, Yelena Rocheva, an admirable woman but no saint; he left a plate of copper to commemorate the event, which was later found). In July, 1841, Fort Ross was abandoned and he spent the remaining time awaiting transport to Sitka (he departed Sept. 5, 1841) at another Russian ranch, the Khlebnikov Ranch inland from the present town of Bodega, probably near present-day Occidental. Details of collection of the specimens are not given, but the specimens are described by Strauch (1870) as

having many relatively large spots (which were thought to be yellowish by Strauch, but in life are whitish). We know that more inland populations have larger spots than coastal populations. We assume the specimens were part of the shipment that accompanied the party as it went to Sitka. Voznesenskii was delayed in making his next shipment to Russia until he returned from a trip to present-day Baja California sometime after March, 1842. This is the shipment that probably reached Russia in 1843.

We think the type specimens were collected in or near the Russian River valley region in the spring of 1841. *Lynch*'s (*1981*) detailed color analysis shows that populations with the most abundant large spots occur inland in northeastern Sonoma and southeastern Mendocino counties. By mid-May salamander activity would have declined, and by June when Mt. St. Helena was climbed Voznesenskii describes the area around Santa Rosa as having the nature of a desert (it does not, but does dry rapidly through May and June, when typically almost no rain falls). The types likely occur within the "Central Core, Clade 2" of *Reilly & Wake* (*2015*, Fig. 3), possibly either within the Lake Berryessa (LB) or the Sonoma (SON) subclades. Both have large, numerous white spots and lie within the Russian River Valley region. Figure 13 shows a series of *A. flavipunctatus* from the Russian River Valley in Sonoma County, and most individuals exhibit large white spots. Figures 12D and 12E show two *A. flavipunctatus* that exhibits spots from just east and west (respectively) of Geyserville in the Russian River Valley. In fact, SON extends to the coast very near Fort Ross (where, however, the spots are relatively subdued). *Storer (1925)* suggests the type locality was in Sonoma County and we agree. Subsequently (as in *Milto & Barabanov, 2011*) "neu-Albion" has been interpreted as "Albion", a coastal village well to the north of Fort Ross, but that place did not exist in 1841 and restriction of the type locality to that site is unwarranted. Both the lectotype (Fig. 14A) and paralectotype (Fig. 14B) have lost their coloration and are thus unable to give us a sense of the number or size of spots.

Of the now four members of the *Aneides flavipunctatus* complex, *A. flavipunctatus* is the most widespread and by far the most internally variable with respect to all variables studied. *Lynch (1981)* showed extensive variation in coloration, coloration ontogeny, morphology, and trunk vertebral numbers which vary from 15 to 18 with means ranging between 16 and 17. *Reilly & Wake (2015)* showed that molecular traits measured also vary substantially and recognized two main clades based on nuclear DNA, in the far northern part of the range and then through the rest of the range to the south and east. Within the northern clade (1) 8 distinct mtDNA clades were identified, with 3 (two of which were relatively highly differentiated internally) in the southern clade (2). In contrast, 3 mtDNA clades were reported in *A. klamathensis* but only one each in *A. iecanus* and *A. niger*. A sharply defined molecular break separates the two clades of *A. flavipunctatus* in the Longvale/Laytonville region of Mendocino Co. where coloration is highly variable (Fig. 15). Future research should focus on this complicated contact zone (*Reilly, Marks & Jennings, 2012*; *Reilly & Wake, 2015*, Fig. 3), where, however, nuclear and mtDNA borders do not coincide. Phylogenomic data are likely needed to fully understand the variation present within *A. flavipunctatus*.

## CONSERVATION CONCERNS

The most vulnerable of the four species of the *Aneides flavipunctatus* complex are *A. niger* and *A. iecanus*. Both have relatively small geographic ranges within which critical habitat has been heavily impacted by humans (discussed below). In contrast, *A. flavipunctatus* and *A. klamathensis* occur over large regions and our recent observations indicate that both are wide-spread within their ranges and that they are locally abundant, with relatively dense populations. We consider them to be of no particular concern from a conservation perspective. Recent survey work has expanded the range limits of this complex and other plethodontid salamanders in the region, and we suggest further surveys of the black salamander complex (e.g., *Olson, 2008*; *Reilly et al., 2010*; *Lindstrand III, Bainbridge & Youngblood, 2012*). Genetically and morphologically distinct populations (especially as we have reported for segments of *A. flavipunctatus)* might best be treated as management units (or ESUs) and periodically monitored and surveyed.

In general, *A. iecanus* is found at elevations below 600 m (*Lynch, 1981*) around Shasta Lake and adjacent areas (as far south as Castle Crags), usually in the vicinity of creeks and in local canyons (Fig. S1). The initial creation of the Shasta Dam and reservoir reduced the range of this species by flooding much of the most suitable habitat in Shasta County. The proposed raising of Shasta Dam would result in the flooding of much of the remaining low elevation habitat around the lake where *A. iecanus* is currently most abundant (Fig. S1). Flooding of this prime habitat and the associated construction/road building activity will have severe negative impacts on this and other sensitive species (e.g., *Hydromantes* web-toed salamanders). Because genetic diversity within this species is low compared to *A. flavipunctatus* and *A. klamathensis* (*Reilly & Wake, 2015*) special attention should be focused on protecting this genetic diversity and maintaining connectivity between genetically distinct populations.

Several workers have recorded that *A. niger* has experienced population declines over the past decades. Members of this once-abundant species have become difficult to find when compared to historical descriptions of its abundance (e.g., *Van Denburgh, 1895*; cf. *Stebbins, 2003*), which may be due to a number of factors including habitat disturbance and destruction (especially along small seeps and creeks), disease (such as pathogenic fungi), and climate change. *Aneides niger* already occurs at the southern extent of the range of the black salamander complex in the Santa Cruz Mountains, which are relatively hotter and drier than the rest of the complex's range. Genetic diversity within this species is also low when compared to *A. flavipunctatus* and *A. klamathensis* (*Reilly & Wake, 2015*) and the protection of genetic diversity within this species should be part of any conservation plan. *Aneides niger* (or *A. f. niger* at the time) is listed as a Priority 3 Species of Special Concern by the California Department of Fish and Wildlife and a detailed treatment of the species can be found in *Thomson, Wright & Shaffer (2016)*.

Additional survey work is critically needed for both *A. niger* and *A. iecanus* to determine their current distribution, estimate actual census population sizes, and determine the level of protection needed to ensure long-term persistence. If it is judged that adequate data concerning these taxa and their biological status exist, we suggest the following categories

of threat using IUCN guidelines (*IUCN, 2012*): *Aneides flavipunctatus* Least Concern (LC)*; Aneides iecanus* Vulnerable (Vu), based on criteria A1, B1b; *Aneides klamathensis* Least Concern (LC)*; Aneides niger* Vulnerable (Vu), based on criteria A1, B1b.

## CONCLUSION

For many years the taxonomy of the *Aneides flavipunctatus* complex has been in a state of flux, with taxonomic proposals ranging from a single species with no subspecies (*Lynch, 1981*) to *Rissler & Apodaca (2007)*, who suggested that four taxa were justified by their data but who took no formal action. For some years we have devoted our efforts to increasing the scope of the study of this complex by adding new data and greatly expanding the geographic extent of the study. We conclude that four taxa are warranted by the data, and have named one new species, raised the rank of one taxon, and removed one taxon from synonymy. However, the remaining *A. flavipunctatus* is a heterogeneous entity that is highly differentiated in all measured traits. Future studies may find justification for additional taxa.

## ACKNOWLEDGEMENTS

We dedicate this paper to the memory of two individuals, both deceased, who conducted their doctoral thesis research on the *Aneides flavipunctatus* complex but came to different conclusions: Charles H. Lowe, Jr and James F. Lynch. We are grateful to Mitchell Mulks who helped with field collections and photography of specimens. We thank Andrew Gottscho, Jon Hirt, Jason Reilly, and many others who helped with the collection of specimens and tissues. SBR thanks Barry Sinervo, W. Bryan Jennings, Sharyn Marks, and Jimmy A. McGuire for their support and help in the development of this study. Lydia Smith and the Evolutionary Genetics Laboratory at UC Berkeley provided laboratory support, Michelle Koo gave GIS support and thoughtful discussions, Sean Rovito helped with morphological analyses, and Carol Spencer facilitated museum accessions and loans.

### Funding

Funding was provided by AmphibiaWeb.org. There was no additional external funding received for this study. The funders had no role in study design, data collection and analysis, decision to publish, or preparation of the manuscript.

### Grant Disclosures

The following grant information was disclosed by the authors:
AmphibiaWeb.org.

### Competing Interests

The authors declare there are no competing interests.

## Author Contributions

- Sean B. Reilly and David B. Wake conceived and designed the experiments, performed the experiments, analyzed the data, contributed reagents/materials/analysis tools, prepared figures and/or tables, authored or reviewed drafts of the paper, approved the final draft.

## Animal Ethics

The following information was supplied relating to ethical approvals (i.e., approving body and any reference numbers):

Animal use was approved by the University of California, Berkeley (IACUC protocol # R093-0205).

## Field Study Permissions

The following information was supplied relating to field study approvals (i.e., approving body and any reference numbers):

Collection of live salamanders in the field was authorized by the California Natural Resources Agency, Department of Fish and Wildlife (#SC-2860 issued to David B. Wake).

## DNA Deposition

The following information was supplied regarding the deposition of DNA sequences:

Raw data is available at GenBank: MK659625–MK659642.

## Data Availability

The raw measurements and museum specimen numbers are available in the Supplemental Files. All specimens and tissues described in the manuscript are stored in the Museum of Vertebrate Zoology at the University of California, Berkeley, USA.

## New Species Registration

The following information was supplied regarding the registration of a newly described species:

*Aneides klamathensis*:
urn:lsid:zoobank.org:act:3A93207C-84B3-4E5A-8A0D-C304C929D539.
*Aneides iecanus*:
urn:lsid:zoobank.org:act:566E1553-669E-40C2-B0D7-E1BB4174873A.
*Aneides niger*:
urn:lsid:zoobank.org:act:EB753380-0F6F-4A26-9867-BB1C33067D4D.
Publication LSID:
urn:lsid:zoobank.org:pub:D11721DC-3000-4EA6-BC51-87D47D3277CB.

## Supplemental Information

Supplemental information for this article can be found online at http://dx.doi.org/10.7717/peerj.7370#supplemental-information.

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
