# Peer review of "Taxonomic revision of black salamanders of the Aneides flavipunctatus complex (Caudata: Plethodontidae)"

_PeerJ, doi:10.7717/peerj.7370_

## Round 0.1 · original submission · Minor Revisions

Dear Authors,

I received two reviews. Both are minor revisions, and I also agree with this assessment. Both reviews are thorough and address all salient points. However, in the revised version I would like to see a short treatment of the conservation status of these species. This could be either associated with each species treatment or before/within the Conclusion section, although a conservation section before/within the Conclusion section is likely to have a greater impact.

I look forward to your revisions.
Sincerely,

Tomas Hrbek

Reviewer 1 ·

Basic reporting

This is an excellent paper overall and should be published once concerns about the multivariate statistical analysis have been addressed (see Experimental design, below). The results clarify the taxonomy of the Aneides flavipunctatus complex to better reflect species-level diversity and evolutionary history. The sampling is impressive and undoubtedly represents thousands of hours of fieldwork and labwork over the years. The manuscript is clearly written, although I urge the authors to consider using the “active voice” in order to improve the strength and impact of the writing. It may not be the authors’ style, but in my opinion would greatly improve the manuscript. I’ve made minor comments and corrections throughout the pdf, which I’ve included. The figures are nice and photos are awesome, but there is a mistake in the legend in Figure 5.

Experimental design

The research question is well-defined and this work clearly fills a knowledge gap. In terms of analysis, the phylogenetic analyses were performed well and support the authors’ conclusions. With respect to the multivariate analysis of morphological data, I have some concerns. I don’t think it makes sense to perform a PCA here. PCA is used as an exploratory method to recover patterns in the data. So, for example, if you didn’t have any pre-defined groups (i.e., you didn’t know the species boundaries from the genetics), you could perform a PCA on the entire complex to see what groupings you found. In this case, however, you already have your 4 well-defined groups based on genetics (species) that you wish to test for differences among. The PCA is unnecessary.

With respect to the discriminant analysis, the canonical portion of the discriminant analysis is appropriate as you’ve performed it. However, the classification function is not (at least as described). The classification function should be validated with a completely separate set of samples, for example, specimens that were not used in creating the canonical function. The “% classified correctly” will always be high (deceptively so) when the prediction is made on the same sample that was used to develop the function. So, I would either take out the classification portion of the discriminant analysis, or perform it correctly by validating it using a resampling procedure (e.g., jackknifing). Alternatively, with a large enough sample, you can use split-sample validation to randomly divide the total sample into two groups, one for analysis, one for validation. I’m not sure your sample size is large enough to do this, however.

Validity of the findings

The findings are valid, though there is one speculative statement (lines 477-478): "...likely insufficient to ever merge these species." I'd take that last bit out. Yes, the genetic estimates suggest a narrow contact zone and low migration, but that's a bit of a leap in my opinion.

Additional comments

This is a great paper, very well done.

Annotated reviews are not available for download in order to protect the identity of reviewers who chose to remain anonymous.

·

Basic reporting

No comment

Experimental design

See my comments and suggestions listed in the General Comments for the Author section.

Validity of the findings

Strengths of the Manuscript:
Taxonomy is often overlooked or relegated to journals specializing in taxonomic descriptions. This paper represents an ideal blend of historical and recent genetic studies all of which point to a conclusion of multiple species of these salamanders. These three new species represent a significant increase to a morphologically and ecologically specialized genus of salamanders with relatively few members making the description important for conservation purposes and to provide a framework for studying diversification. The authors were able to weave in the discussion of species relationships and other important considerations within the species descriptions themselves making the descriptions a key part of understanding the evolution of these taxa.

The historical facts discussed are an important part of both the species descriptions and key to understanding the species names and previous ideas regarding their status. I have included some suggestions regarding some of the historical details below.

The writing is excellent and clear with the exceptions noted in comments for the author.

Additional comments

Review of Taxonomic revision of black salamanders of the 1 Aneides flavipunctatus complex (Caudata:Plethodontidae) (#36410) by Sean B. Reilly and David B. Wake:

Points by importance, then in order:
1) For the morphological analyses, I assume that all of the specimens listed (the paratypes and holotypes) constitute the points in Figure 4 and the numbers in Table 2. If so, this should be made more explicit both in the methods and materials and in the results. In addition, being able to identify the individuals present in the contact zone in Figure 2 would help to either show whether individuals close the contact zone are distinct in morphometric space, or not. I suggest circling or numbering the individuals in Figure 4 that are from populations listed in Figure 2. Perhaps a subscript number with the population numbers from figure 2.
For table 2, it would be useful to know if the misidentified specimens were in the contact zone, or not.
In addition, I would like to see the distinction (or lack thereof) morphometrically (specifically referring to Figure 4 and Table 2) in the contact zone discussed somewhere in the manuscript. If animals close to the contact zone are more distinct ten elsewhere, this raises the possibility of reproductive character displacement and/or reinforcement.
2) Figure 5
Aneides flavipunctatus is mislabeled as Aneides niger (repeated twice on the figure)
3) Figure 3 - The red coloration of the new samples is confusing as they appear to indicate Aneides klamathensis sequences because that species is indicated with red - I suggest a symbol like an asterisk to indicate new samples
The figure legend should state clearly that there is no mitochondrial introgression between geographic localities.
Within the box, “Populations 7-12” in the upper box and 1-6 in the lower box would help to clarify the distinct nature of the populations.

Lines 63-64 This sentence should be re-worded to make it clearer that Storer considered Aneides iecanus synonymous with Aneides flavipuctatus and therefore considered all previous references to Plethodon iecanus, Aneides iecanus and Autodax iecanus to all be considered Aneides flavipunctatus.

From Storer 1925:
Plethodon flavipunctatus Strauch (1870, pp. 71-72). Original description, type from New Albion [probably the coastal portion of Sonoma County], California.
Plethodon flavipunctatus, Boulenger (1882b, pp. 55-56).
Plethodon iecanus Cope (1883, pp. 24-25). Type locality, Baird, Shasta
County, California.
Aneides iecanus, Cope (1886, p. 526). Generic allocation.
Plethodon iecanus, Townsend (1887, pp. 240-241).
Plethodon flavipunctatus, Cope (1889, p. 145).
Autodax iecanus, Cope (1889, pp. 187-189, text fig. 46). General account. Autodax iecanus, Van Denburgh (1895b, pp. 776-778). Range; breeding
habits; eggs.
Autodax iecanus, Ritter and Miller (1899, p. 696).
Plethodon flavipunctatus, Yan Denburgh (1916, p. 221).
Aneides iecanus, Grinnell and Camp (1917, pp. 135-136, fig. 2). Range. Aneides iecanus, Stejneger and Barbour (1917, p. 21; 1923, p. 18). Range.

Cope had moved Plethodon iecanus to Aneides iecanus in 1886 (see above) - perhaps this should be added to the history of the taxonomic changes.

Line 441 -"manuscript name" does that mean a new name in the Lowe manuscript?

Lines 604-605 Did Cope move iecanus to Aneides in 1886 on page 576? see above comment in Storer 1925:
From Cope 1886:
Notes. — I add here that the Plethodon iecanus Cope proves to be a well- marked species of Anaides. The species was described from a young one.

Cope 1886. Synonymic list of the North American species of Bufo and Rana, with
descriptions of some new species of Batrachia, from specimens in
the National Museum. Proc. Am. Philos. Soc., 23, 514-526.

Line 653-654:
Ieka was recognized as the word for Mt. Shasta in a few other sources:

A History of Northern California; A Memorial and Biographical History
Chicago, IL: Lewis Publishing Co., 1891.
Page 241: "...the tribe of lndians that inhabit Scott and Shasta valleys and the mountains to tbe north. The true name of their tribe they have forgotten or will not tell, having been called Shasta for half a century; but the name of their beautiful patron mountain still remains to us, Ieka, the white."

Also, the town of Yreka was reported to have been originally Ieka --
Snopes:
https://www.snopes.com/fact-check/yreka-bakery/
An 1876 article from the Yreka Journal quoted on Snopes.com:
"It was intended the county seat should bear the name of Ieka, the Indian name of Mount Shasta, but by mistake the name of Wyreka was substituted and the error continued, with the exception of dropping the letter W, thought to be superfluous."

Figure 10
The spots on this specimen as well as living specimens I can recall look blueish - is this incorrect? Is it worth mentioning that the small spotting looks blue from a distance, but is white when examined close up?

---

## Round 0.2 · accepted · Accept

Dear Authors,

Thank you for the revisions. I have read through your responses, and your updated MS, and I am thoroughly satisfied. So I am happy to accept your revised MS. You have done a very nice job on resolving some of the long-standing taxonomic issue of this very interesting group of salamanders.

Congratulations on a job well done.

Sincerely,

Tomas Hrbek